# Biodiversity mediates the effects of stressors but not nutrients on litter decomposition

**Léa Beaumelle[1,2†]\*, Frederik De Laender[3], Nico Eisenhauer[1,2]**

[1]German Centre for Integrative Biodiversity Research (iDiv) Halle-Jena-Leipzig, Leipzig, Germany; [2]Institute of Biology, Leipzig University, Leipzig, Germany; [3]Research Unit of Environmental and Evolutionary Biology, Namur Institute of Complex Systems, and Institute of Life, Earth, and the Environment, University of Namur, Namur, Belgium

**Abstract** Understanding the consequences of ongoing biodiversity changes for ecosystems is a pressing challenge. Controlled biodiversity-ecosystem function experiments with random biodiversity loss scenarios have demonstrated that more diverse communities usually provide higher levels of ecosystem functioning. However, it is not clear if these results predict the ecosystem consequences of environmental changes that cause non-random alterations in biodiversity and community composition. We synthesized 69 independent studies reporting 660 observations of the impacts of two pervasive drivers of global change (chemical stressors and nutrient enrichment) on animal and microbial decomposer diversity and litter decomposition. Using meta-analysis and structural equation modeling, we show that declines in decomposer diversity and abundance explain reduced litter decomposition in response to stressors but not to nutrients. While chemical stressors generally reduced biodiversity and ecosystem functioning, detrimental effects of nutrients occurred only at high levels of nutrient inputs. Thus, more intense environmental change does not always result in stronger responses, illustrating the complexity of ecosystem consequences of biodiversity change. Overall, these findings provide strong evidence that the consequences of observed biodiversity change for ecosystems depend on the kind of environmental change, and are especially significant when human activities decrease biodiversity.

**\*For correspondence:**
lea.beaumelle@gmail.com

**Present address:** †INRAE, UMR SAVE, Villenave d'Ornon, France

**Competing interests:** The authors declare that no competing interests exist.

## Introduction

Human activities cause global environmental changes with important consequences for biodiversity and the functioning of ecosystems. Understanding these consequences is crucial for better policy and conservation strategies, which will ultimately promote human well-being too (**IPBES, 2019**). A key question is to what extent changes in ecosystem functioning are mediated by changes at which dimensions of biodiversity. Extensive research has demonstrated that biodiversity is needed for the stable provenance and enhancement of ecosystem processes and functions (**Cardinale et al., 2012**; **Schuldt et al., 2018**; **Tilman et al., 2012**). However, this body of evidence is mostly based on experiments comparing ecosystem functioning in artificial communities with varying number of species. Such experiments might not capture the complex ways by which shifts in biodiversity induced by global change ultimately affect ecosystem functioning (**De Laender et al., 2016**; **Eisenhauer et al., 2019b**).

Early biodiversity-ecosystem function (BEF) experiments typically controlled for environmental gradients, thus not accounting for the underlying drivers of biodiversity change (**De Laender et al., 2016**; **Srivastava and Vellend, 2005**; **Wardle, 2016**). These early experiments also focused on species richness as the sole biodiversity index, and manipulated it directly and randomly. However,

**eLife digest** Ecosystems are at their healthiest when they have a high level of biodiversity – that is, a wide variety of different species all living together. But human activity is changing the environment and putting ecosystems under pressure. One of the places this is most evident is in the communities of organisms responsible for breaking down dead plants.

These organisms – called decomposers – are highly sensitive to pesticides, metals and other chemical stressors, as well as excess nutrients, such as nitrogen, released by industry and farming. Exposing decomposers to these chemicals can change both the number of individuals of each species and the number of different species that are present. In other words, these chemicals can, respectively, alter both the abundance and diversity of decomposer communities. Controlled experiments in simplified conditions suggest that these changes in biodiversity affect ecosystem health. But, it remained unclear to what extent these results applied to real-world scenarios of environmental change.

To test the findings of controlled experiments, Beaumelle et al. investigated how chemical stressors and excess nutrients affect the breakdown of leaf litter – the debris of decomposing leaves that forms on top of soil. Previous studies suggest that the reduced biodiversity caused by chemicals should result in leaf litter breaking down more slowly. Whereas the loss in biodiversity caused by nutrients will increase the number of some decomposer species, causing leaf litter to break down faster or slower, depending on local conditions.

Beaumelle et al. tested these predictions by gathering the results from 69 independent studies conducted across the globe. The results showed that stressors caused the diversity and abundance of decomposers to decline, which reduced the breakdown of leaf litter, as expected. But, the outcomes of excess nutrients were more varied. Low levels of excess nutrients increased the breakdown of leaf litter, but at high levels slowed down the rate leaves decomposed. Furthermore, the effect excess nutrients had on biodiversity in decomposer communities changed according to the types of organisms in the ecosystem. This suggests that variations in biodiversity can impact ecosystems differently depending on the type of environmental change.

The breakdown of leaf litter plays a critical role in carbon balance, and this has knock-on effects for the Earth's climate. This work suggests that improving biodiversity is crucial to maintain the health of ecosystems, but successful strategies will have to be adjusted depending on the type of human impact (for example, chemical stressors or nutrient additions). These findings could help researchers design better approaches for boosting ecosystem health in the future.

environmental change will often elicit non-random changes in several facets of biodiversity (*Eisenhauer et al., 2016*; *Giling et al., 2019*; *van der Plas, 2019*) (community composition and population densities (*Glassman et al., 2018*; *Spaak et al., 2017*), functional diversity (*Cadotte et al., 2011*; *Craven et al., 2018*; *Heemsbergen et al., 2004*), trophic diversity (*Soliveres et al., 2016*; *Wang and Brose, 2018*; *Zhao et al., 2019*). The selective effects of environmental change emerge because organisms differ in their response to environmental change. For example, larger organisms and predators are often more negatively affected than smaller organisms at lower trophic levels (*Hines et al., 2015*; *Sheridan and Bickford, 2011*; *Srivastava and Vellend, 2005*; *Voigt et al., 2007*). Using realistic extinction scenarios, experiments found contrasting effects of non-random shifts in biodiversity on ecosystem functioning (e.g. *Cárdenas et al., 2017*; *Jonsson et al., 2002*; *Melguizo-Ruiz et al., 2020*; *Oliveira et al., 2019*; *Smith and Knapp, 2003*, *Zavaleta and Hulvey, 2004*). In addition, several variables that are not directly related to biodiversity control ecosystem functions (e.g. physiological rates [*Dib et al., 2020*; *Thakur et al., 2018*] and alterations of physical and chemical conditions [*De Laender et al., 2016*; *Giling et al., 2019*]). When environmental change affects these mechanisms, teasing out the relative importance of biodiversity-mediated effects is complicated even more. Given the number of different potential mechanisms, quantifying the extent to which shifts in biodiversity underpin the effect of environmental change on ecosystem functioning under real-world scenarios of global change is a key challenge for ecology (*De Laender et al., 2016*; *Duffy et al., 2017*; *Eisenhauer et al., 2019b*; *Srivastava and Vellend, 2005*; *van der Plas, 2019*; *Wardle, 2016*). Incorporating the impacts of environmental change drivers into BEF studies and

meta-analyses is an important step forward to address such questions (*De Laender et al., 2016*; *Eisenhauer et al., 2019b*).

The vast majority of BEF experiments has focused on plant richness and ecosystem functions such as biomass production (*van der Plas, 2019*). However, litter decomposition has a tremendous importance in ecosystems and biogeochemical cycles (*Follstad Shah et al., 2017*). Small changes in the rate of this process can have important consequences for the overall carbon balance. Indeed, increases in decomposition rates could have positive feedback effects on climate warming by enhancing C losses (*Kirschbaum, 2000*). The diversity of decomposers (invertebrates and microorganisms that fragment and decompose organic matter in both aquatic and terrestrial systems) is crucial for litter decomposition (*Eisenhauer et al., 2012*; *García-Palacios et al., 2013*; *Gessner et al., 2010*; *Handa et al., 2014*; *Hättenschwiler et al., 2005*) and for other ecosystem functions as well (*Eisenhauer et al., 2019a*; *Lefcheck et al., 2015*; *Schuldt et al., 2018*). Despite the importance of decomposers, BEF experiments focusing on litter decomposition more often addressed the influence of plant litter diversity than of decomposers (*Gessner et al., 2010*; *Tonin et al., 2018*). In a meta-analysis, decomposer diversity had a greater effect on decomposition than the diversity of plant litter (*Srivastava et al., 2009*), although also weak and neutral effects have been reported (*van der Plas, 2019*). Facilitation and complementarity through niche partitioning are primary mechanisms underlying the positive relationship between decomposer diversity and decomposition (*Gessner et al., 2010*; *Hättenschwiler et al., 2005*; *Tonin et al., 2018*). Experiments conducted in natural conditions and reflecting realistic extinction scenarios are still relatively scarce, and demonstrate contrasting effects of non-random shifts in decomposer diversity on decomposition (*Cárdenas et al., 2017*; *Jonsson et al., 2002*; *Melguizo-Ruiz et al., 2020*; *Wenisch et al., 2017*). The need to quantify environmental change effects on decomposer diversity, along with potential knock-on effects on litter decomposition, is therefore particularly pressing.

There is a variety of environmental change drivers, and different types of drivers may have diverse effects on biodiversity and ecosystem functions (*De Laender et al., 2016*; *Dib et al., 2020*). We postulate that there are two main categories of environmental change: stressors and resource shifts. While stressors cannot be consumed, and act as conditions that alter growth rates (e.g., temperature, drought, chemical stressors), resources are by definition consumed (e.g., $CO_2$ or mineral nutrients), which has important implications for how they should enter theory (*Chase and Leibold, 2003*; *De Laender, 2018*). Chemical stressors and nutrient enrichment are important case studies of environmental stressors and resource enrichment, because their presence is increasing rapidly (*Bernhardt et al., 2017*) and they are projected to have severe effects on biodiversity (*Mazor et al., 2018*). They are also of particular relevance for decomposer communities. Chemical stressors such as metals and pesticides decrease the diversity, abundance, growth and activity of decomposers across terrestrial and aquatic systems (e.g. *Hogsden and Harding, 2012*; *Pelosi et al., 2014*; *Schäfer, 2019*). In contrast, nutrient enrichment can have positive impacts on the abundance and physiological rates of decomposer organisms by reducing resource limitations (*Treseder, 2008*), but at the same time decrease decomposer diversity (*Lecerf and Chauvet, 2008*; *Woodward et al., 2012*). Across ecosystems, stressors and nutrients can exert opposite impacts on litter decomposition rates, with decreases in response to chemical stressors but increases following nutrient enrichment (*Ferreira et al., 2015*; *Ferreira et al., 2016*). In addition, decomposition involves both microorganisms and invertebrates (*Bardgett and van der Putten, 2014*; *Gessner et al., 2010*; *Hättenschwiler et al., 2005*) that may respond differently to stressors and nutrients with a higher sensitivity of invertebrates than microorganisms (*Peters et al., 2013*; *Siebert et al., 2019*). Although many published case studies report shifts in decomposer diversity and in rates of litter decomposition at sites impacted by stressors and nutrients, biodiversity-mediated effects have not yet been quantified across systems.

Here we addressed the question if the effects of stressors and nutrient enrichment on decomposer diversity and abundance explain the response of litter decomposition to these two types of pervasive environmental change drivers (*Figure 1*). We synthesized 69 published case studies reporting the impact of stressors (metals, pesticides) and nutrients (nitrogen or phosphorous additions) on litter decomposition and on decomposer diversity (taxa richness, Shannon diversity, evenness) or abundance (density, biomass) at sites differing in stressor or nutrient levels. Our comprehensive global dataset of 660 observations encompasses studies across taxonomic groups (animal (soil micro-, meso- and macrofauna, stream macroinvertebrates) and microbial (fungi and bacteria)

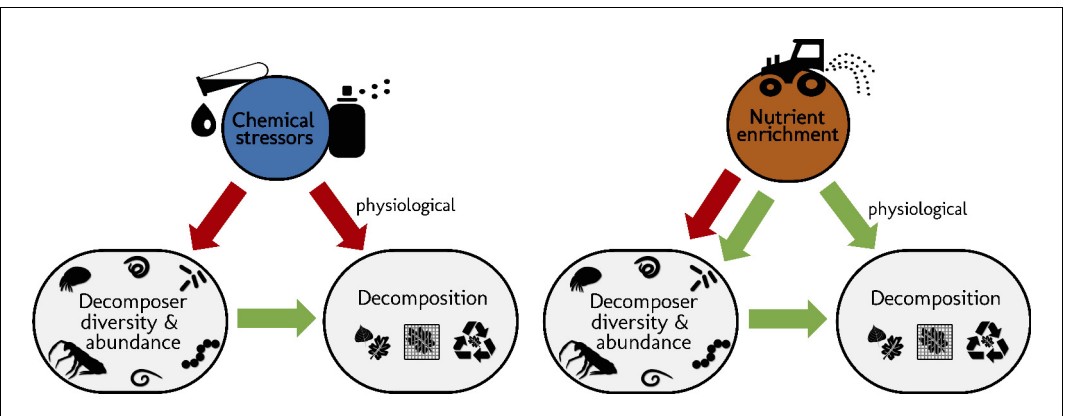

**Figure 1.** Schematic representation of the structural hypotheses tested in this study. Green arrows depict expected positive effects, red arrows represent negative effects. Stressors and nutrients are hypothesized to decrease decomposer diversity. The response of decomposers diversity to environmental change drivers determines the response of decomposition (*Srivastava et al., 2009*). Nutrients are hypothesized to increase decomposer abundance. Stressors and nutrients can affect litter decomposition independent of changes in decomposer diversity and abundance, especially through changes in physiological activity (*De Laender et al., 2016*; *Giling et al., 2019*).

decomposers), ecosystems (aquatic and terrestrial), and study types (experimental and observational) (*Figure 2*). We quantified the effect size of environmental change on decomposer diversity or abundance and on litter decomposition within studies using correlation coefficients between stressor or nutrient levels and decomposer diversity, abundance, and litter decomposition. We also characterized stressor and nutrient intensities, and standardized their levels in water, soil, or sediment using environmental quality criteria issued by environmental authorities (e.g. ECHA, USEPA, UKTAG). Using meta-analysis and structural equation modelling (SEM), we first compared the overall effects of stressors and nutrients on decomposers and decomposition across systems and studies (first meta-analysis), and second, addressed to what extent changes in decomposer diversity and abundance mediate the impacts of these two contrasting drivers of environmental change on decomposition (second meta-analysis and SEM). Third, we explored the effects of three main moderators on decomposers diversity, abundance, and decomposition responses, as found in the second meta-analysis: stressor or nutrient intensity, taxonomic group (animal vs. microbes) and study type (experimental vs. observational studies).

We expected that chemical stressors and nutrients would have contrasting effects on decomposer diversity and abundance, and on litter decomposition across ecosystems and studies (*Figure 1*). We hypothesized that chemical stressors generally decrease decomposer diversity, abundance (*Hogsden and Harding, 2012*; *Petrin et al., 2008*), and litter decomposition rates (*Ferreira et al., 2016*; *Peters et al., 2013*), and that nutrients generally decrease decomposer diversity (*Lecerf and Chauvet, 2008*; *Woodward et al., 2012*) but increase decomposer abundance and litter decomposition rates (based on physiological effects and decreasing resource limitations (*Bergfur et al., 2007*; *Ferreira et al., 2015*; *Treseder, 2008*; *Woodward et al., 2012*). We further hypothesized that litter decomposition responses to environmental change depend on changes in decomposer diversity and abundance, and expected an overall positive relationship independent of environmental change intensity (*Srivastava et al., 2009*).

## Results

### Description of the data and overall patterns

The final dataset contained 69 (case) studies from 59 publications, representing 660 observations. Data were mostly from Europe (44 ; 443 (studies; observations)) and North and South America (19; 168), while Asia (2; 9) and Oceania (4; 40) were less well represented (*Figure 2A*). The studies covered aquatic (55; 388) and terrestrial systems (14; 272) (*Figure 2C*), and used observational (43; 336)

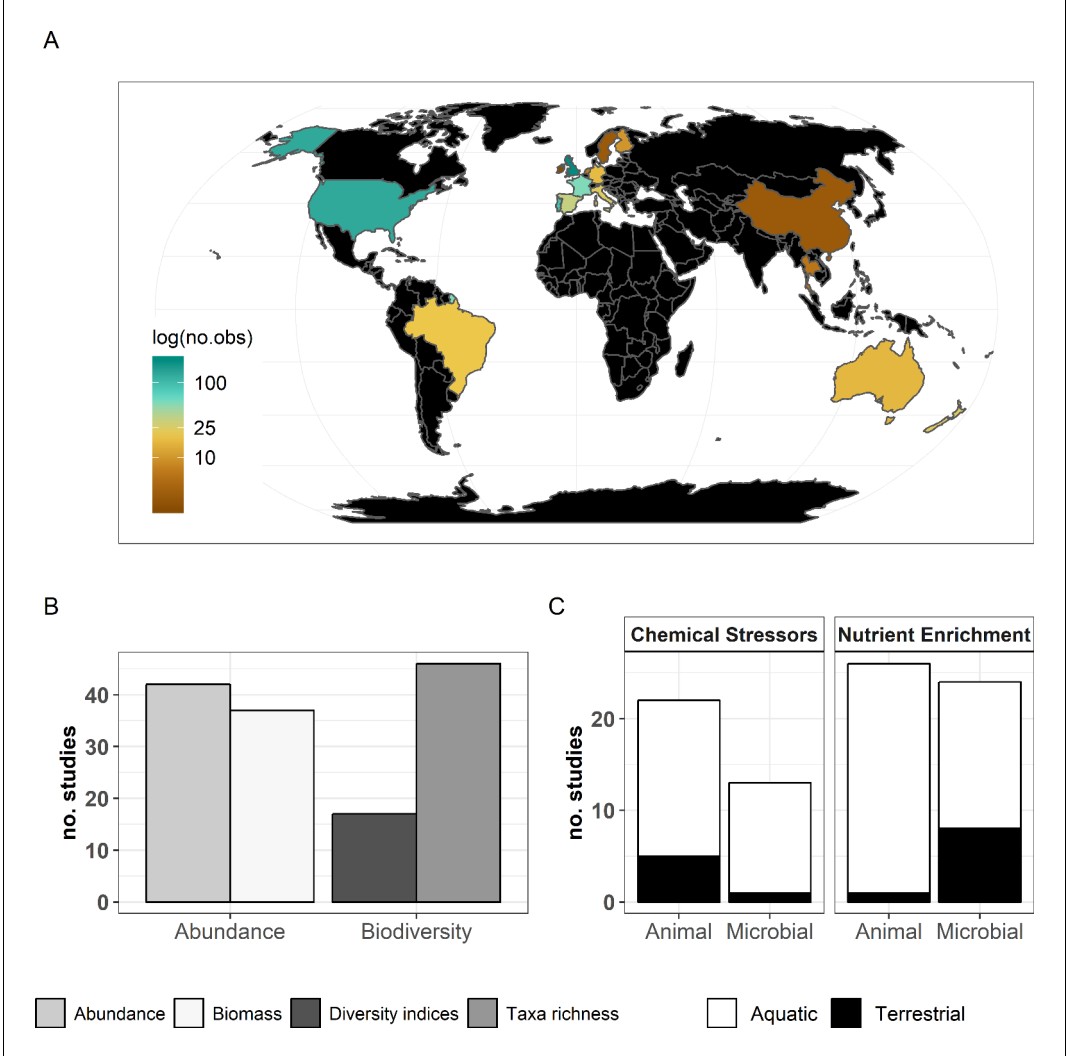

**Figure 2.** Description of the data used in the present meta-analysis. (**A**) Countries represented and corresponding number of observations, (**B**) decomposer diversity and abundance metrics covered, and (**C**) ecosystem types and decomposer taxonomic groups (animals: soil micro-, meso-, macro-fauna, stream macroinvertebrates; and microbial decomposers: fungi and bacteria) represented.

or experimental approaches (26; 324). Studies reported abundance (66; 463) or diversity responses (48; 197) (*Figure 2B*) of soil and benthic invertebrates (48; 509) and microbes (fungi and bacteria) associated with litter materials (36; 151) (*Figure 2C*). Chemical stressors were mostly metals (13; 257) and pesticides (12; 66) associated with industrial activities, accidental spills, and agricultural practices. Nutrient enrichment studies addressed fertilization by various N and/or P forms (26; 175), and eutrophication due to agricultural runoffs (10; 59) or wastewater effluents (4; 44). There was no study reporting nutrient enrichment impacts on soil decomposer diversity in the dataset. Funnel plots and intercepts of Egger's regression showed evidence for positive publication bias in nutrient enrichment studies reporting decomposer abundance (*Appendix 2—figure 1*; *Appendix 2—figure 2*; *Appendix 2—table 1*). No publication bias was detected in the other datasets.

We found largely contrasting effects of stressors and nutrients on each of the three response variables in a first-level meta-analysis comparing the overall effects of the two drivers of environmental change (*Figure 3*, *Appendix 2—table 2*). Chemical stressors overall decreased decomposer diversity, abundance and litter decomposition across studies (*Figure 3*). Nutrient enrichment tended to decrease decomposer diversity but to increase abundance, and decomposition, although these

trends were not significant as indicated by confidence intervals of the grand mean effects overlapping with zero (**Figure 3**).

## Biodiversity-mediated effects of stressors and nutrients on litter decomposition

The responses of decomposition and of decomposer diversity and abundance to chemical stressors were correlated: decreases in decomposition were associated with decreases in decomposer diversity and abundance (**Figure 4** upper panels). We did not find such a relationship for nutrients. Instead, a range of positive and negative responses of decomposer diversity, abundance, and decomposition to nutrients were found, without significant associations between them (**Figure 4** lower panels). In addition, when decomposer diversity and abundance responses to nutrients were close to zero, there was a wide range of decomposition responses (intercepts from **Figure 4** lower panels).

According to our overarching hypothesis, the SEM indicated that the effects of stressors on litter decomposition were mediated by shifts in decomposer diversity and abundance. Including the direct paths from decomposer diversity or abundance to litter decomposition improved both the models according to mediation tests and AIC comparisons (**Figure 5**). In addition, the path coefficients from diversity and abundance to the decomposition response to stressors had (standardized) values higher than 0.1 (**Figure 5**) and were statistically different from zero (**Appendix 2—table 3**). However, in contrast to chemical stressors, the SEM did not support biodiversity-mediated effects of nutrient enrichment on litter decomposition. While the mediation test and AIC indicated that the decomposer diversity-mediated path improved the model (**Figure 5**), the path coefficient was not significantly different from 0 (**Appendix 2—table 3**). The decomposer abundance-mediated path of nutrients was not supported by the data: an SEM without the direct path from decomposer abundance to decomposition could not be rejected based on the mediation test (**Figure 5**), and including this path did not improve the model according to the AIC comparison. Besides, we found publication bias in this dataset (**Appendix 2—figure 2**, **Appendix 2—table 1**), and model check indicated that the residuals of the nutrients-abundance model were non-independent from the fitted values. Thus, the results from this model are reported here for comparison purposes only.

The magnitude of the biodiversity-mediated effects of chemical stressors on decomposition was stronger than that of the direct effects of stressor intensity on decomposition. The indirect effect of stressors on decomposition mediated by diversity (i.e. mathematical product of the standardized paths from stressor intensity to decomposer diversity and from diversity to decomposition **Figure 5**) was higher than the direct effect of stressors on decomposition, while the abundance-mediated effect of stressors was negligible (**Figure 5**). In the case of nutrient enrichment, however, decomposition responses were not explained by shifts in decomposer diversity and abundance, and the direct effects of nutrient intensity dominated the total effect (**Figure 5**). Finally, between-model comparisons (based on unstandardized path coefficients [**Grace, 2006**]) revealed that decomposer diversity

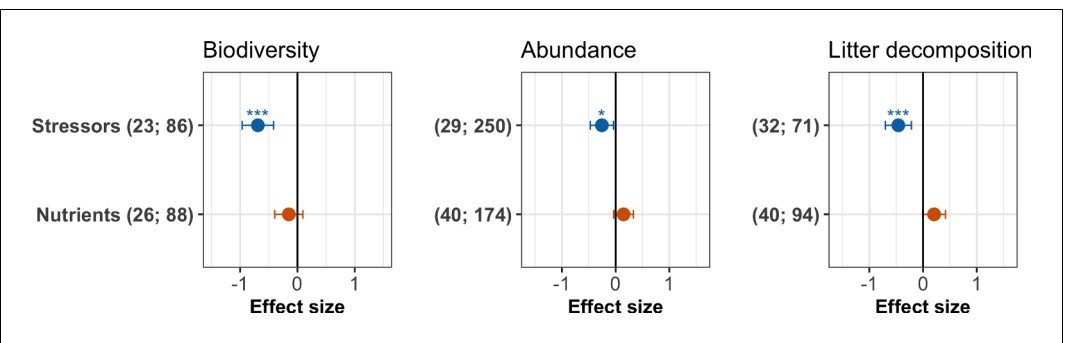

**Figure 3.** Grand mean effect sizes of chemical stressors and nutrient enrichment on decomposer diversity (taxa richness and diversity indices), abundance (density and biomass), and litter decomposition. Effect sizes are z-transformed correlation coefficients. Error bars show 95% confidence intervals. Numbers in parentheses indicate number of studies and observations, respectively. Symbols show the significance level for the comparison between mean effect size and zero (***p<0.001; *p<0.05). For full model results, see **Appendix 2—table 2**.

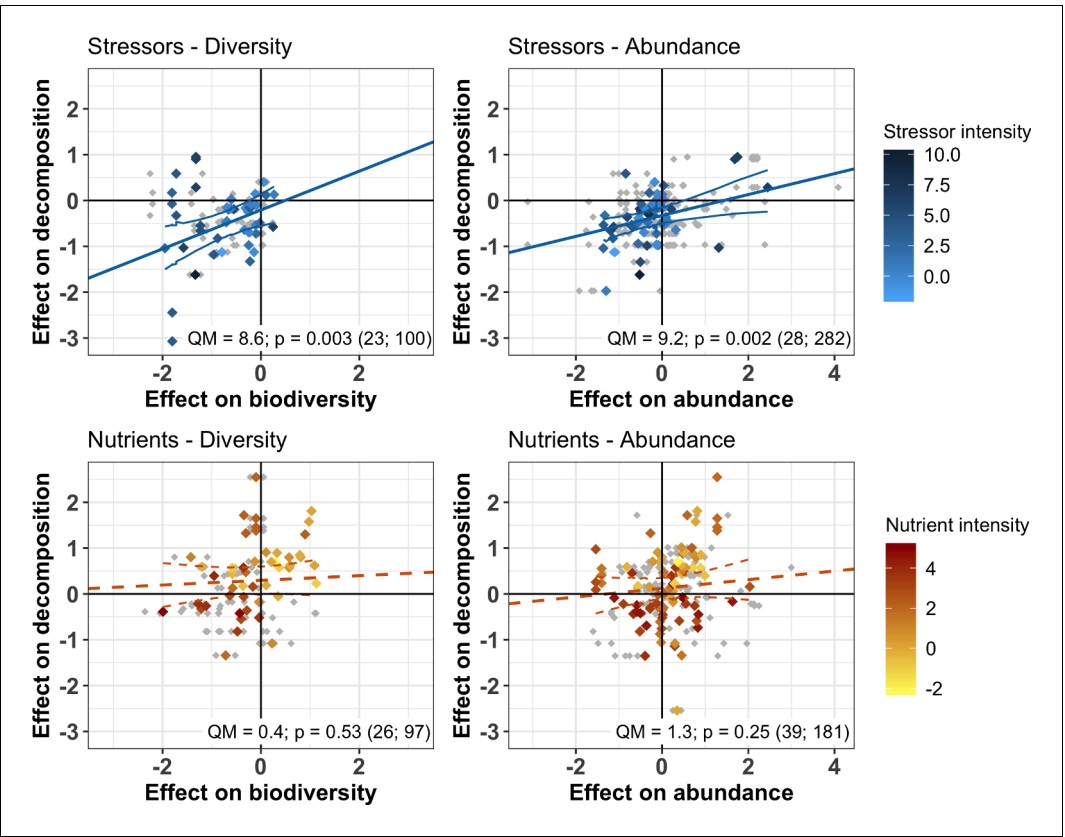

**Figure 4.** Relationship between the responses of decomposition and decomposer diversity and abundance to chemical stressors and nutrient enrichment. Variables are effect sizes (z-transformed correlation coefficients) of stressors or nutrients on litter decomposition and on animal and microbial decomposer diversity (left panels) or abundance/biomass (right panels). Gray symbols are individual observations of effect sizes; Colored symbols indicate mean effect size on diversity or abundance across individual observations for a unique litter decomposition measurement (see methods). Lines represent meta-regressions between effect sizes for decomposition and decomposers, where solid lines are statistically significant (p<0.05), dashed lines are non-significant (p>0.05), and thin lines depict the regression's confidence interval. QM and p represent the model heterogeneity and p-values of the meta-regressions, respectively, with sample size (number of studies; number of observations).

was a stronger driver of decomposition response to stressors than decomposer abundance (unstandardized paths were 0.42 and 0.24 for diversity and abundance, respectively, *Appendix 2—table 3*).

Sensitivity analyses revealed that the results were robust to the inclusion of approximated standard deviations (*Appendix 3—table 1*; *Appendix 3—table 2*), and extreme values of effect sizes (*Appendix 3—table 3*; *Appendix 3—table 4*). We found partially different results when using log-response ratios as effect sizes (*Appendix 3—table 5*; *Appendix 3—table 6*), due to lower sample sizes and emergence of extreme values in these datasets. In addition, the log-response ratio is probably sensitive to the various metrics of biodiversity, abundance, and decomposition covered by the individual studies that we included, while correlation coefficients better accommodate such discrepancies (*Koricheva et al., 2013*).

## Response of animal and microbial decomposers and decomposition to stressor and nutrient intensity

Despite the overall negative effects of stressors on decomposition, negative responses in decomposition were not associated with higher stressor intensity (*Figure 5*, *Figure 6*). This result held for two complementary approaches: multivariate SEM (*Figure 5*) that relied on data resampling to account for replicated values of decomposition matching several decomposer responses (e.g. for different taxa in the same litterbag), and meta-regressions (*Figure 6*) where data resampling was not

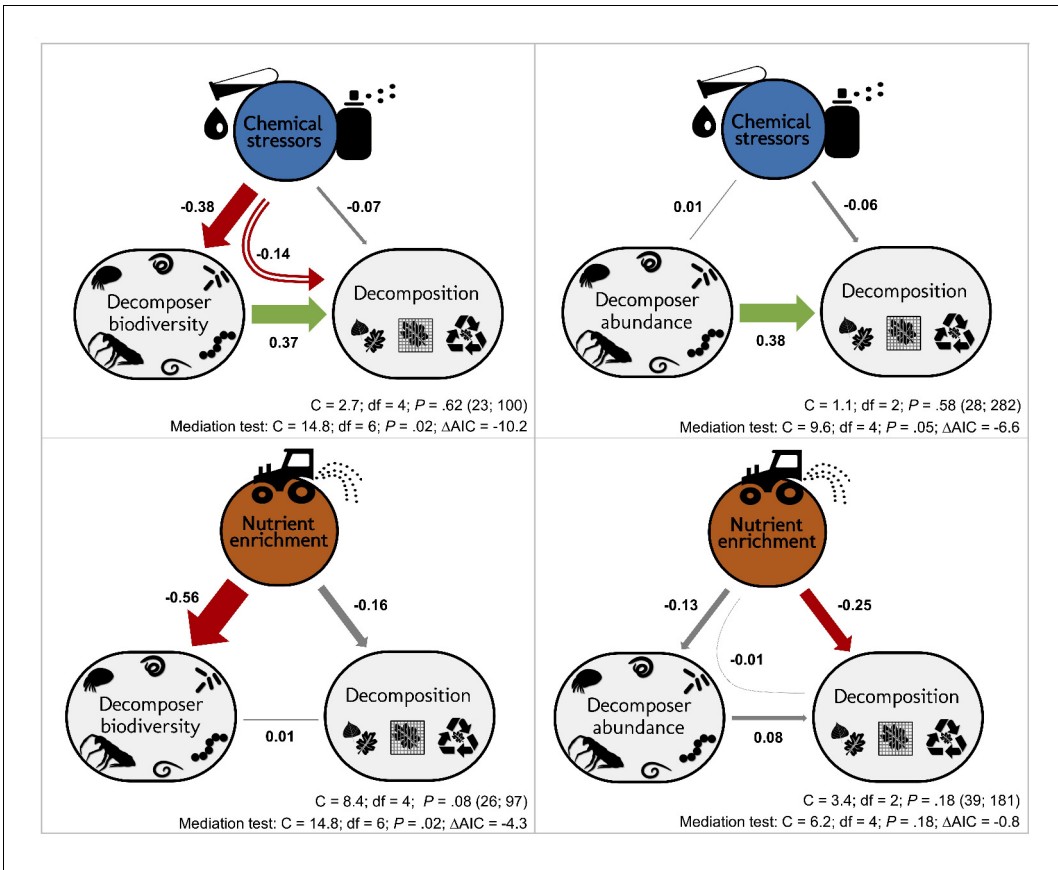

**Figure 5.** Decomposer diversity and abundance explained litter decomposition response to chemical stressors but not to nutrient enrichment. Structural equation models investigating decomposer diversity- or abundance-mediated effects of chemical stressors and nutrient enrichment on litter decomposition across 69 studies. Arrows represent relationships between stressor or nutrient intensity levels, and effect sizes of stressors or nutrients on litter decomposition and on decomposer diversity (taxa richness, Shannon diversity, or evenness: left panels) or abundance and biomass (right panels). Values along the arrows are standardized path coefficients. Green, red, and gray arrows indicate positive, negative, and non-significant relationships, respectively. Curved arrows depict the indirect effects of stressors or nutrients on decomposition as mediated by diversity or abundance. Arrow widths are scaled relative to the magnitude of standardized path coefficients. C statistic, $P$-value ($P<0.05$ indicate poor model fit), and sample sizes (number of studies; number of observations). Results of mediation tests: comparison with models omitting the path from diversity or abundance to decomposition ($\Delta$AIC $< -2$ indicates that reduced models were not consistent with the data).

necessary (see Materials and methods). There was mixed support for a stressor intensity effect on decomposer diversity across the two approaches: decomposer diversity responses decreased with stressor intensity according to the SEM (*Figure 5*), but this trend was not significant according to the second level meta-analysis (*Figure 6*). Similar slopes were obtained both with the SEM relying on data resampling (the slope of the relationship was $-0.10 \pm 0.04$, *Appendix 2—table 3*) and with the meta-regression (the slope was $-0.05 \pm 0.03$). The differences between the two approaches can be explained by the different data included. Decomposer abundance responses were not associated to stressor intensity in both the SEM and meta-regression approaches (*Figure 5*, *Figure 6*). We found different patterns for nutrient enrichment, where decomposition responses decreased with nutrient intensity (*Figure 5*, *Figure 6*), from positive effects at low intensity to negative effects at higher intensity (*Figure 6*). A similar pattern was observed for decomposer diversity, where responses decreased with nutrient intensity from positive to neutral to negative responses at high nutrient levels (*Figure 6*). Nutrient intensity, however, did not explain the responses of decomposer abundance (*Figure 5*, *Figure 6*), and both positive and negative responses were found at high nutrient levels.

The meta-analysis further revealed clear discrepancies between the response of animal and microbial (fungi and bacteria) decomposers to stressors and nutrients. Animal decomposers responded more strongly to chemical stressors than microbial decomposers. The mean effects of chemical stressors on animal decomposer diversity and abundance were more negative than that on microbial decomposers, confirmed by Wald type tests of the second-level meta-analyses (*Figure 7* upper panels, *Appendix 2—table 4*). Animal decomposers overall decreased in diversity but increased in abundance in response to nutrient enrichment (*Figure 7*, lower panels). On the other hand, the mean effects of nutrients on microbial decomposer diversity and abundance had lower magnitudes compared to animals (*Appendix 2—table 4*), with confidence intervals overlapping with zero (*Figure 7* lower left panel). Finally, there was no clear difference between observational and experimental studies (*Figure 7*, *Appendix 2—table 4*), and between biodiversity responses in terms of taxa richness or of diversity indices (*Appendix 2—table 4*).

## Discussion

The present synthesis brings new insights into how changes in decomposer biodiversity induced by two pervasive drivers of environmental change ultimately affect decomposition. We find concomitant changes in biodiversity and decomposition under the influence of chemical stressors but not nutrient enrichment, highlighting that real-world patterns relating shifts in biodiversity and ecosystem functioning depend on the type of environmental change. In fact, we observed significant correlations between effects on biodiversity and ecosystem function in a scenario where chemical stressors caused a significant decline in biodiversity. In contrast, in cases where nutrient enrichment caused variable responses in biodiversity, relationships between biodiversity and ecosystem function responses were weaker. It remains an understudied but important question if results of controlled BEF experiments are applicable to non-random changes in biodiversity caused by human activities (e.g. *De Laender et al., 2016*; *Duffy et al., 2017*; *Eisenhauer et al., 2019b*; *Srivastava and Vellend, 2005*; *van der Plas, 2019*; *Wardle, 2016*). The present results provide strong empirical evidence for significant real-world BEF relationships when environmental changes decrease biodiversity.

### Biodiversity-mediated effects of chemical stressors on decomposition

Chemical stressors caused consistent reductions in decomposer diversity and abundance as well as in litter decomposition rates, in line with several previous case studies (*Beketov et al., 2013*; *Malaj et al., 2014*) and meta-analyses (*Ferreira et al., 2016*; *Peters et al., 2013*). Adding to the previous knowledge, the present meta-analysis shows that changes in decomposer diversity and abundance explained the decomposition response to stressors, providing evidence for the expectation that shifts in biodiversity mediate the impact of chemical stressors on decomposition. We acknowledge that despite the SEM analysis, the approach conducted here remains correlative. However, our study builds on a body of experimental and observational evidence that already demonstrated that more diverse and abundant decomposer communities support higher decomposition rates, albeit not under the influence of environmental change (e.g. *García-Palacios et al., 2013*; *Handa et al., 2014*).

We especially complement a previous meta-analysis showing the importance of decomposer diversity for decomposition across experiments manipulating the richness of invertebrate and microbial decomposer communities (*Srivastava et al., 2009*). We extend on this and show that non-random biodiversity losses induced by stressors are closely associated with decreases in decomposition across a wide range of studies. A recent review pointed out that in naturally assembled terrestrial communities, studies more often found neutral and to a lesser extent positive relationships between decomposer diversity and decomposition (*van der Plas, 2019*). In that review, communities were not influenced by environmental change drivers, and the vote counting approach used is sensitive to the statistical power of individual studies and could have increased the probability of finding non-significant relationships (*Koricheva et al., 2013*). In line with our findings, an experiment mimicking the sequence in which freshwater invertebrate decomposers are lost after disturbances showed that decreasing non-randomly the number of species decreased decomposition rates (*Jonsson et al., 2002*).

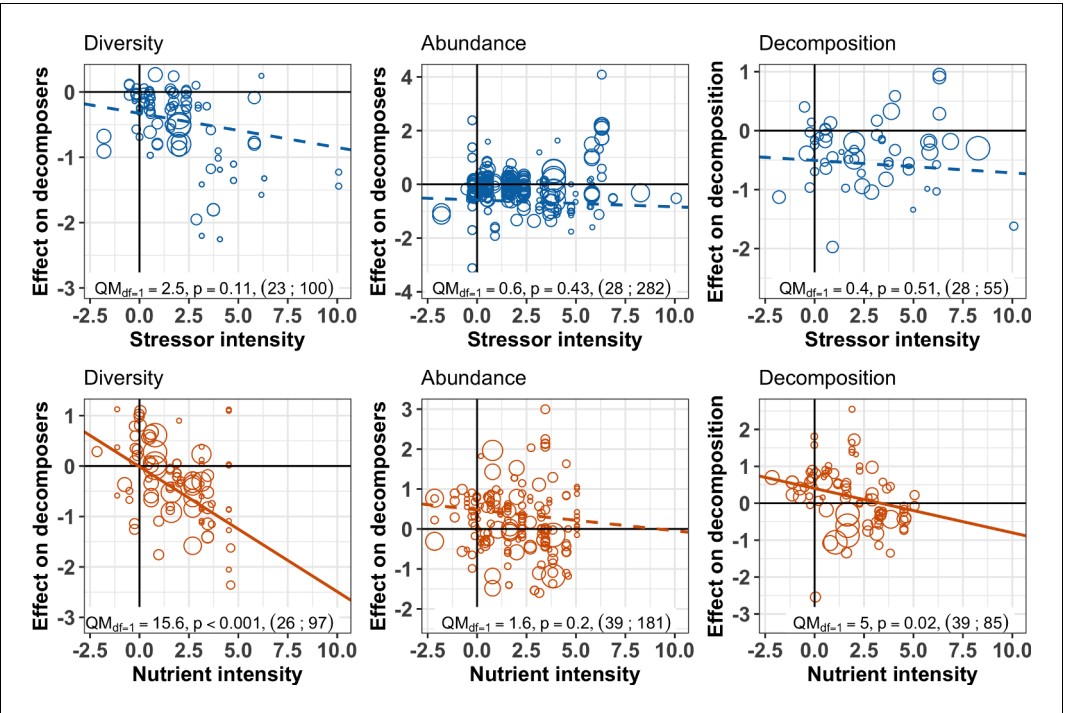

**Figure 6.** Decomposer and decomposition responses to the intensity levels of chemical stressors and nutrient enrichment. Values are effect sizes (z-transformed correlation coefficients). Stressor or nutrient intensity represents the standardized level of environmental change in the treatment with the highest level (values < 0: observed level below quality criteria considered to be safe for the environment; values > 0: observed level above quality criteria). Point size is proportional to the inverse of the variance in effect size. Lines are the slopes and 95% confidence intervals from bivariate meta-regressions, with associated QM statistics, p-value and sample size (number of studies; number of observations).

Biodiversity-ecosystem function experiments manipulating biodiversity directly are key to understand the mechanisms involved in this relationship (*Eisenhauer et al., 2016*), especially because they control for the effects of environmental heterogeneity or abundance. However, in real-world scenarios, environmental change drivers affect both biodiversity and abundance simultaneously. As demonstrated here, this is especially the case for stressors that decrease decomposer diversity and abundance (*Hogsden and Harding, 2012*). The abundance or biomass of different decomposers is of critical importance for decomposition (e.g. *Bergfur et al., 2007*; *Ebeling et al., 2014*; *Manning and Cutler, 2018*). Even at constant richness and community composition, strong decreases in abundance can have important impacts on ecosystem functioning (*Spaak et al., 2017*; but see *Dainese et al., 2019*). It is beyond the scope of the present meta-analysis to disentangle the effects of biodiversity from the effects of abundance, and we found that both contributed to explain shifts in decomposition in separate analyses. It is interesting to note that the few cases where negative effect sizes of stressors on biodiversity were associated with positive effect sizes on decomposition were also cases where decomposer abundance was positively associated with stressors (*Figure 4*). Although we cannot specifically test this with the present data, it seems that in those particular cases (*Lucisine et al., 2015*), increases in decomposer abundance counteracted the negative effects of decreases in decomposer diversity (*Dornelas et al., 2019*). Those results could therefore be in line with the mass-ratio hypothesis (*Grime, 1998*; *Smith and Knapp, 2003*). Indeed, an exclusion experiment showed that dominant, small, detritivores can compensate reductions in litter decomposition caused by the removal of large detritivores (*Cárdenas et al., 2017*). These concomitant shifts in both diversity and abundance further have important implications for our estimates of diversity responses, as studies mostly reported richness to estimate decomposer diversity, but rarely corrected for the sampling effort (*Gotelli and Colwell, 2001*). This means that lower abundances rather than a lower number of species per se might have directly caused some of the negative

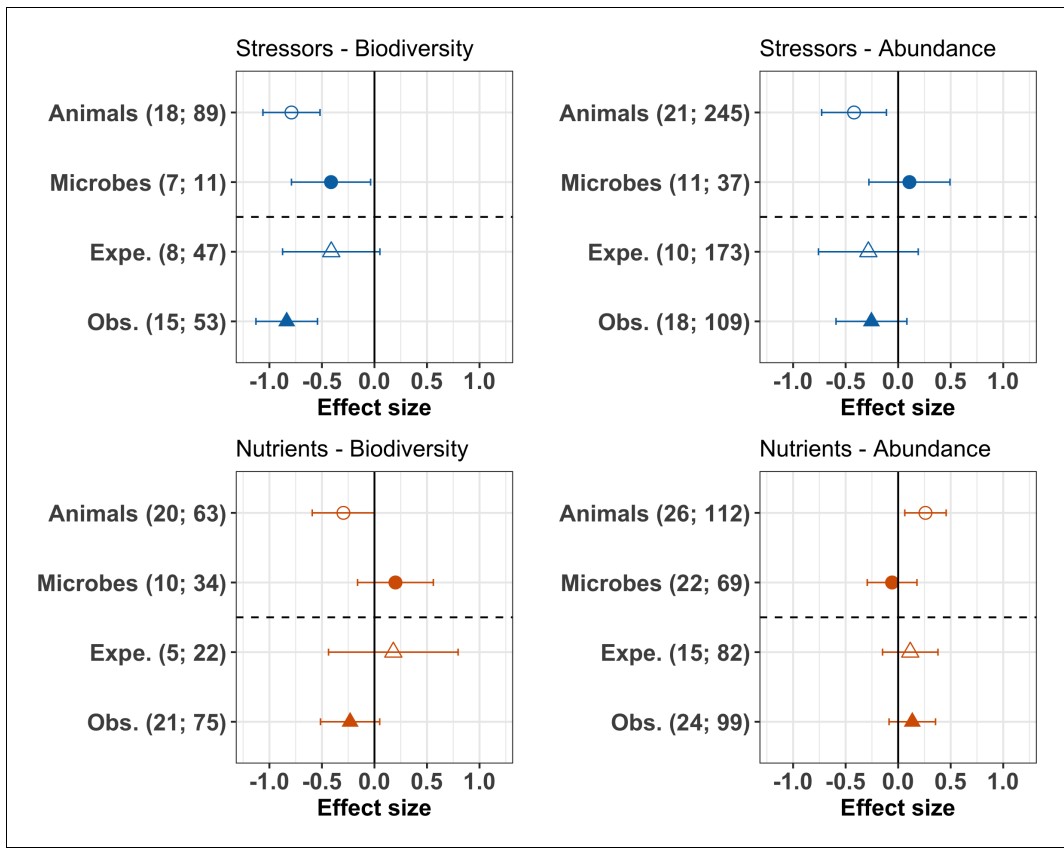

**Figure 7.** Moderator effects on decomposer diversity and abundance responses to chemical stressors and nutrient enrichment. Responses of decomposer diversity (taxa richness and diversity indices) and abundance (densities and biomass) to stressors and nutrients according to the taxonomic group (animals and microbes) and study type (Expe. = experimental; Obs. = observational studies). Values are mean effect sizes (z-transformed correlation coefficients) and 95% confidence intervals derived from meta-analytic models. Sample sizes are reported for each moderator: (number of studies; number of observations).

effects on biodiversity reported here (*Chase and Knight, 2013*). This common caveat in meta-analysis approaches that rely on how individual studies report biodiversity, also applies to the present study, and reinforces the importance of reporting raw data in future studies on the impacts of chemical stressors on biodiversity and ecosystem functioning.

The effects of changes in decomposer diversity and abundance on decomposition found in the present study might also have channeled changes in community and food-web structure not captured by our biodiversity metrics. Changes in keystone species (*Hättenschwiler et al., 2005*), functional diversity (*Cadotte et al., 2011*; *Dangles et al., 2012*; *Heemsbergen et al., 2004*), vertical diversity (*Gessner et al., 2010*; *Melguizo-Ruiz et al., 2020*; *Wang and Brose, 2018*; *Zhao et al., 2019*), or dominance patterns (*Dangles and Malmqvist, 2004*) might have shifted concomitantly to taxonomic diversity and abundance. Moreover, these different components of diversity might act at different timings of decomposition (*Oliveira et al., 2019*). Unfortunately, studies rarely reported such measurements together with decomposition. For example in our dataset, only seven studies reported evenness. Future studies need to explore shifts in decomposer community composition in more detail to better understand what particular aspect of biodiversity is responsible for changes in decomposition rates (*Giling et al., 2019*; *Hättenschwiler et al., 2005*). In particular, few of the included studies reported comparable functional groups allowing to address the effect of functional diversity across the multiple systems and taxonomic groups addressed by the present analysis. Future synthesis work could specifically address the effect of functional diversity, by focusing on a given system type. Indeed, there is ample evidence that shifts in functional diversity are crucial for decomposition (*Heemsbergen et al., 2004*), and that facilitative interactions occur primarily

between decomposers of contrasting body size (*Dangles et al., 2012*; *Tonin et al., 2018*). This is especially the case for interactions between animal and microbial decomposers, where fragmentation of litter by detritivores facilitates access for microbial decomposers (*Eisenhauer et al., 2010*; *Hättenschwiler et al., 2005*; *Yang et al., 2012*).

Here, we found that invertebrates were more affected by chemical stressors than microbes, across aquatic and terrestrial ecosystems. Invertebrate decomposers are particularly sensitive to the impacts of metals and pesticides (*Hogsden and Harding, 2012*; *Pelosi et al., 2014*; *Peters et al., 2013*; *Schäfer, 2019*). Microbial decomposers are known to be sensitive to metals (*Giller et al., 2009*) and pesticides as well (*DeLorenzo et al., 2001*). Nevertheless, our result is consistent with the general expectation that larger organisms are more sensitive to environmental change due to longer generation time, higher energetic demands and lower population densities (*Hines et al., 2015*; *Sheridan and Bickford, 2011*; *Woodward et al., 2005*; *Yvon-Durocher et al., 2011*; *Baas and Kooijman, 2015*). These different sensitivities between groups of decomposers could imply that the biodiversity-mediated effects of stressors on decomposition are more strongly linked to shifts in invertebrates than microbes, as reported in a previous review (*Peters et al., 2013*). However, in another meta-analysis focusing on microbial-driven decomposition rates, changes in fungal biomass and richness explained shifts in decomposition under the impacts of chemical stressors, but also of nutrient enrichment (*Lecerf and Chauvet, 2008*).

## Nutrient-induced changes in decomposition were not related to shifts in decomposer diversity

The impacts of nutrient enrichment on litter decomposition and decomposer diversity were different from those caused by stressors, confirming our expectations. These different biodiversity and function responses led to different emergent relationships between decomposer diversity and decomposition compared to stressors. We found that nutrients had a variety of effects ranging from positive to negative depending on the taxonomic group (*Figure 7*) and nutrient intensity (*Figure 6*), and resulting in neutral overall mean effects (*Figure 3*). Previous syntheses also found positive (*Ferreira et al., 2015*) as well as inconsistent (*Knorr et al., 2005*) responses of decomposition rates to nutrient enrichment in streams. The relatively small mean effect of nutrient enrichment on decomposition in the present meta-analysis could be explained by the use of correlation as an effect size, which does not capture potentially non-monotonic responses of decomposition to nutrients (*Woodward et al., 2012*). However, we noted that most of the studies included in the present meta-analysis did not individually span nutrient gradients sufficiently large to capture this potential non-monotonous response. Taken together, the studies show positive effects on decomposition at low nutrient intensities that shifted toward neutral to negative effects at higher intensities (*Figure 6*), which is consistent with previous findings (*Ferreira et al., 2015*; *Woodward et al., 2012*). Low-nutrient intensities might have enhanced microbial activity and biomass by alleviating resource limitation, resulting in enhanced decomposition. At higher intensities, however, negative impacts on invertebrates might have decreased decomposition rates (*Peters et al., 2013*; *Woodward et al., 2012*).

These nutrient intensity patterns contrasted with the results for chemical stressors. The overall negative effects of stressors (*Figure 3*) on decomposition were not explained by stressor intensity levels (*Figure 6*), and there was mixed support for a stressor intensity effect on decomposer diversity based on two complementary data analysis approaches (SEM based on data resampling (*Figure 5*) vs. second level meta-analysis *Figure 6*). Thus, negative responses to chemical stressors happened across the range of stressor intensity. Such contrasting patterns between stressor and nutrient intensity effects may reflect the greater number of stressor types (different metals, pesticides, mixtures) covered by individual studies compared to the limited number of nutrients. In addition, due to the higher variability of stressor types, we relied on more variable sources to standardize stressor levels compared to nutrients in the diversity dataset (Materials and methods, *Appendix 1—table 1*). With the data at hand, it was not possible to test the influence of the environmental quality criteria used to standardize stressor and nutrient levels, because such an effect would be confounded with stressor or nutrient types. The datasets were all dominated by environmental quality criteria based on similar methodologies (for 75% to 100% of observations, see Material and Methods). However, future studies focusing on stressor intensity effects across ecosystems would greatly benefit from coordinated efforts to derive quality criteria encompassing the vast and rapidly increasing number of chemical stressors (*Wang et al., 2020*).

Contrary to our expectation, nutrient-induced shifts in decomposer diversity and abundance were not associated with shifts in decomposition rates across studies. We found that increasing nutrient intensity decreased the effects on decomposition and on decomposer diversity, but not on decomposer abundance. Statistically controlling for the effect of nutrient intensity with SEM indicated no residual association between shifts in decomposer diversity or abundance and in decomposition rates, that is a non-significant BEF relationship. Changes in microbial abundance in response to nitrogen deposition explained the responses of different ecosystem functions in terrestrial systems in previous meta-analyses (*García-Palacios et al., 2015*; *Treseder, 2008*). Here, we show that this pattern cannot be generalized across aquatic and terrestrial systems and across animal and microbial decomposers. Contrary to stressors, when the diversity and abundance of animal and microbial decomposers were not affected by nutrients, we observed large positive and negative shifts in decomposition (intercepts of *Figure 4*), that were explained by nutrient intensity (*Figure 4*: negative effects on decomposition at invariant biodiversity are associated with high intensities and positive effects with lower intensities). Together, these results show that nutrient-induced shifts in decomposer diversity were not as strong drivers of decomposition changes as stressor-induced biodiversity shifts. These differences may be partly due to the different mechanisms underlying the effects of stressors and nutrients. Based on previous studies, we speculate that our results are due to the complex responses of animal and microbial decomposers at different nutrient intensities (*Ferreira et al., 2015*; *Lecerf and Chauvet, 2008*; *Treseder, 2008*; *Woodward et al., 2012*).

Animal decomposers showed a stronger response to nutrients than microbes. Invertebrate decomposers overall decreased in diversity, but they increased in abundance under nutrient enrichment. These results could reflect a loss of sensitive taxa to the benefit of tolerant taxa that were able to use additional resources and would then increase in density (*Bergfur et al., 2007*). Overall, microbial decomposers responded little to nutrient enrichment, probably reflecting a mixture of positive and negative effects that nutrients can have on microbial growth (*Lecerf and Chauvet, 2008*; *Treseder, 2008*), as well as on different microbial taxa. Indeed, nutrients can alleviate resource limitations at low intensities, but can also exert toxic effects at high intensities. The initial levels of nutrients thus condition subsequent responses in decomposers and decomposition to nutrient enrichment (*Ferreira et al., 2015*; *Knorr et al., 2005*). Furthermore, at high intensities, nutrients can be associated with other chemical stressors (e.g. pesticides in agricultural runoffs) (*Ferreira et al., 2015*; *Woodward et al., 2012*). The influence of interactive effects of stressors and nutrients was impossible to quantify with the data at hand, given that only a few experiments assessed the effects of both drivers independently, but many observational studies may have been confounded by such joint effects. Chemical stressors and nutrients are often co-occurring in e.g. agricultural landscapes, and the consequences of such combinations are still poorly understood (*Alexander et al., 2013*; *Alexander et al., 2016*; *Barmentlo et al., 2018*; *Chará-Serna et al., 2019*; *Chará-Serna and Richardson, 2018*; *Fernández et al., 2016*). Furthermore, stressor and nutrient effects might be modulated by climatic and other environmental conditions, and studies on interaction effects are scarce (*Rillig et al., 2019*; *Thakur et al., 2018*). Finally, although our comparison of stressors versus resources allowed us to test a clear concept, any kind of grouping in ecological studies may mask some of the variation within the categories and future studies may be interested in different categories. Indeed, a given environmental change driver can represent a stressor for a given species, and a resource for another species (*Connell et al., 2018*). As data availability improves, future work could include different environmental change drivers. This would also allow to test additional groupings of drivers and ecological concepts unifying stressors and resources (*De Laender, 2018*; *Harley et al., 2017*).

## Conclusions

This study brings new insights into the real-world patterns relating ecosystem function to non-random changes in biodiversity induced by environmental change. We found that the consequences of changes in biodiversity for ecosystem functioning depend on the type of environmental change. Real-world scenarios do not necessarily involve concomitant changes in both biodiversity and function across terrestrial and aquatic systems. We further found that with the environmental quality criteria used in risk assessment, there were already significant positive and negative effects on decomposers and decomposition (*Figure 6*), highlighting the need to better incorporate biodiversity and ecosystem function into ecological risk assessment programs (*De Laender and Janssen, 2013*).

Finally, we report overall negative effects of chemical stressors on biodiversity and ecosystem functioning across terrestrial and aquatic ecosystems that reinforce recent calls to consider chemical stressors as important global change drivers and address their impacts on biodiversity and ecosystems (*Bernhardt et al., 2017*; *Mazor et al., 2018*; *Steffen et al., 2015*). Positive real-world BEF relationships may be particularly significant in cases where environmental changes decrease biodiversity, such as in the case of chemical stressors. Such information are crucial if we are to design policy and conservation strategies able to reconcile human development with biodiversity conservation.

## Materials and methods

### Data collection

We searched the Web of Science for studies that addressed the impact of environmental drivers and recorded decomposer community responses and litter decomposition rates. The search strategy is fully reported in Supplementary Methods (Appendix 1). The search retrieved 2536 references. Abstracts and titles were screened to identify a final set of 61 records that met our inclusion criteria (PRISMA plot, *Appendix 1—figure 1*, and list of included references (Appendix 4). To be included in the meta-analysis, studies had to:

- Report litter decomposition (rates, mass loss, proportion of mass remaining) and the diversity, abundance, or biomass of decomposers at sites differing in chemical stressor or nutrient levels.
- Focus on naturally assembled communities subjected to the impact of chemical stressors or nutrient enrichment. Studies that manipulated decomposer diversity directly were not considered to only focus on non-random biodiversity change scenarios. We included mesocosm studies only when they used field-sampled communities and left time for the community to reach an equilibrium in mesocosms in order to reflect real-world conditions as much as possible.
- Report the response of animal (benthic macroinvertebrates, or soil micro, meso or macrofauna) or microbial decomposers (bacteria or fungi from decomposing leaves or in surrounding water or soil samples).
- Report decomposer abundance (density or biomass), or decomposer diversity (taxa richness, Shannon diversity, evenness).

When a reference reported different environmental change drivers or geographical areas with a specific reference site for each case, we considered these as individual (case) studies (*García-Palacios et al., 2015*). We extracted means or sums, standard deviations, and sample sizes of litter decomposition, decomposer diversity, and abundance (outcomes) in non-impacted vs. impacted sites (control-treatment studies), or at each site when gradients of chemical stressors or nutrients were investigated (gradient studies). When response variables were reported at different time points, we kept only the last time point to capture long-term responses. For studies reporting decomposition, decomposer abundance or diversity for several litter types (e.g. different litter species), several groups of organisms (e.g. functional feeding groups for macroinvertebrates), and several diversity metrics (e.g. Shannon indices and taxon richness), we created separate observations within case studies. We also extracted chemical stressor or nutrient levels at those sites (water, soil, or sediment concentrations of chemical stressors or nutrients, or application rate of pesticides or fertilizers). The study type (experimental vs. observational), taxonomic group (animal decomposers or microbial decomposers) and metric of diversity (taxa richness or diversity indices (Shannon diversity and evenness)) were also recorded. We used the online software Webplotdigitizer to extract data from figures (*Rohatgi, 2018*). We converted standard errors and confidence intervals into standard deviations using the equations in *Lajeunesse, 2013*. When reported as mass loss, litter decomposition data were transformed into k rates using the exponential decay equation used in *Ferreira et al., 2015*.

### Effect size calculation

We used z-transformed correlation coefficients as effect sizes in order to cope with the heterogeneity of data and study types (*Koricheva et al., 2013*). For control-treatment studies, we first calculated Hedge's d, and then transformed Hedge's d into correlation coefficients (*Lajeunesse, 2013*). For gradient studies (four or more treatment levels), we calculated correlation coefficients between

the mean values of abundance, diversity, or decomposition rate and the corresponding chemical stressor or nutrient concentrations. When means, standard deviations, or sample sizes were missing, we contacted the authors to retrieve the data. When the information could not be retrieved, standard deviations were approximated from the data, using the linear relationship between mean values and standard deviations across our datasets (*Lajeunesse, 2013*).

## Standardization of chemical stressors and nutrient enrichment intensities

Given the variability in the different stressors and nutrients combinations in the studies, stressor and nutrient levels were standardized into a common environmental change driver intensity ($ECD_{intensity}$) as follows:

$$ECD_{intensity} = log([Compound_i]_{treatment} / [Compound_i]_{criteria})$$

where $[Compound_i]_{criteria}$ were environmental quality criteria set by European or US environmental authorities for the chemical stressor or nutrient considered (*Appendix 1—table 1*), and $[Compound_i]_{treatment}$ were the concentrations of the chemical stressor or nutrient at the treatment or impacted sites. When multiple stressors or nutrients were reported, we used the standardized intensity of the stressor or nutrient corresponding to the highest standardized intensity for the rest of the analyses.

We used consistent sources for the environmental quality criteria as much as possible. For chemicals, we relied primarily on quality criteria from the European Chemical Agency (ECHA) and United States Environmental Protection Agency (USEPA) that use standardized procedures across aquatic and terrestrial realms based on ecotoxicological data. For nutrients, we relied mostly on European Water Framework Directive (WFD) benchmarks. Using various sources for those quality criteria was inevitable due to the high number of chemicals and the various way the authors reported stressor or nutrient levels in individual studies. When we could not find quality criteria for the stressors or nutrients considered in the studies in our main sources, we relied on the authors' statements and expert knowledge regarding their stressor or nutrient levels (e.g. citation for ecotoxicological data, or synthesis studies, or recommended application rates of pesticides [*Appendix 1—table 1*]). Despite this, the final datasets were all dominated by similar sources for standardizing stressor and nutrient intensity levels: thresholds from ECHA or USEPA for 80% and 90% of observations in the stressor-diversity and stressor-abundance datasets, respectively, and for nutrients, thresholds from WFD for 100% and 75% of observations in the nutrient-diversity and nutrient-abundance datasets, respectively.

## Overall effects of chemical stressors and nutrient enrichment: first-level meta-analysis

We first tested the differences between the effects of chemical stressors and nutrient enrichment on decomposer diversity, abundance and litter decomposition responses by quantifying the grand mean effect sizes on the three response variables (first level meta-analysis). Three separate meta-analyses were conducted, one for each response variable, and included the type of driver (stressors or nutrients) as a categorical moderator, and a random effect of the case study. We used a weighted meta-analysis giving more weight to effect sizes derived from studies with larger sample sizes. Weights were the inverse of the variance in z-transformed correlation coefficients (*Viechtbauer, 2010*). Publication bias was evaluated using funnel plots with environmental change driver type as covariate. The intercepts from Egger's regressions (standardized effect size vs. precision = 1/SE) were inspected for significant deviation from zero that would indicate publication bias (*Koricheva et al., 2013*). Residual plots were used to detect strong deviation from normality and outliers. We estimated the grand mean effect sizes and compared the effect of chemical stressors and of nutrients using Wald-type chi-square tests. The `rma.mv()` function of the R package metafor was used (*R Development Core Team, 2018*; *Viechtbauer, 2010*).

## Relationship between biodiversity and decomposition: Structural equation modelling

An SEM was fitted to estimate the relationship between decomposer diversity or abundance and litter decomposition responses to environmental change drivers while controlling for the joint influence of stressor or nutrient intensity and categorical covariates. We used piecewise SEM (*Lefcheck, 2001*) estimating two linear mixed effect models, one for decomposition ($z_{LD}$) and one for decomposer diversity or abundance responses ($z_B$), with a random effect of the case study on the intercepts. These two sub-models embedded in the piecewise SEM were the second-level meta-analyses in our hierarchical approach. The random effect structure, weighting approach and variance structure were coded with the R package nlme (*Pinheiro et al., 2018*) in a way that fully reproduced the meta-analysis approach of weighting and of known residual variance (*Viechtbauer, 2017*):

$$z_{LD} \sim z_B + ECD_{intensity} + study\,type, random = \sim 1|Case\,study/ID$$

$$z_B \sim ECD_{intensity} + study\,type + taxonomic\,group\,(+diversity\,metric), random = \sim 1|Case\,study/ID$$

This SEM was tested separately for each of four datasets: Stressors – Biodiversity; Stressors – Abundance; Nutrients – Biodiversity and Nutrients – Abundance datasets. The influence of the diversity metric (diversity indices versus taxa richness) was tested in the Biodiversity datasets only. We initially considered more complex model structures, but were unable to use them for analysis due to data limitations (in particular the effect of the ecosystem type and of interactions between our covariates).

Outliers, relationships between covariates, and non-linear patterns between continuous covariates were explored graphically. Studies often reported different decomposer diversity or abundance values for the same litter decomposition (e.g. when several taxonomic or functional groups were reported in the same litterbag). This variability could have affected the model estimates. We thus used data resampling to account for duplicated effect sizes on litter decomposition in the analyses. A stratified resampling was conducted, where for each duplicated value of effect size on decomposition, one randomly selected effect size on biodiversity was kept at each out of 1000 iterations. The models were fitted for each data resampling iteration, and we averaged model estimates and statistics across iterations and used the means as final values (path coefficients and standard error of the path and intercepts, Chi-square statistics and AICs).

Goodness-of-fit of the SEMs was assessed using directed separation tests based on the Fisher's C statistic. We used mediation tests to explore the significance of the path between decomposer diversity or abundance and litter decomposition based on the Fisher's C statistic of SEM that did not include the biodiversity-mediated path (*Lefcheck, 2001*; *Shipley, 2009*). We calculated the p-value associated with the mean Fisher's C statistic across data resampling iterations (p-value<0.05 indicated poor model fit). The AICs of models with and without the biodiversity-mediated paths were further compared using averaged AICs across data resampling iterations. We considered the biodiversity (or abundance) path to be consistent with the data when the SEM without the biodiversity-path had p-value<0.05 (poor fit) and was not associated with a better AIC value (i.e. lower than two units) than the SEM including the biodiversity path. Residuals from the two sub-models of each SEM were graphically evaluated for strong departure to normality and relationship with the fitted values (*Duffy et al., 2015*). For these analyses, we averaged the residuals across data resampling iterations for each observation. We finally compared the relative magnitude of the biodiversity-mediated path versus the direct path from stressor or nutrient intensity to litter decomposition based on the mathematical product of the standardized path coefficients (*Grace, 2006*).

## Moderator analyses: second-level meta-analyses

In order to quantify the influence of the categorical (study type, taxonomic group and diversity metrics) and continuous (environmental change intensity) moderators on the three response variables, we further analyzed the results of the second-level meta-analyses (i.e. the sub-models embedded in the SEMs). The data resampling used in the SEM was no longer necessary, because there were no repeated values of decomposition matching different decomposer diversity or abundance measurements in this univariate approach. We quantified the effects of the different moderators based on the Wald-type chi-square tests derived with the R package metafor (*Viechtbauer, 2010*).

## Sensitivity analyses

We finally tested the robustness of the results to the approximation of standard deviations, the presence of extreme values, and the metric of effect size used. The analyses were re-run with datasets that did not include the effect sizes for which we approximated standard deviations, for datasets that did not include extreme values of effect sizes (values beyond the whiskers of boxplots that is below quantile 1 minus 1.5 times the interquartile range or above quantile 3 plus 1.5 times the interquartile range). Finally, we calculated log-response ratios instead of correlation coefficients as effect sizes and re-run the analyses.

## Acknowledgements

LB was funded by the Synthesis Centre (sDiv) of the German Centre for Integrative Biodiversity Research (iDiv) Halle-Jena-Leipzig, funded by the German Research Foundation (FZT 118). We are grateful to Helen R P Phillips, Benjamin Rosenbaum, Adam T Clark and Katharina Gerstner for data analysis advice and to Simone Cesarz for creating the images for *Figures 1* and *5*.

## Additional information

### Funding

| Funder | Grant reference number | Author |
|---|---|---|
| Synthesis Centre (sDiv) of the German Centre for Integrative Biodiversity Research (iDiv) Halle-Jena-Leipzig, funded by the German Research Foundation | FZT 118 | Léa Beaumelle |

The funders had no role in study design, data collection and interpretation, or the decision to submit the work for publication.

### Author contributions

Léa Beaumelle, Conceptualization, Data curation, Formal analysis, Funding acquisition, Investigation, Visualization, Methodology, Writing - original draft, Writing - review and editing; Frederik De Laender, Nico Eisenhauer, Conceptualization, Supervision, Funding acquisition, Writing - review and editing

### Author ORCIDs

Léa Beaumelle [iD] https://orcid.org/0000-0002-7836-8767
Nico Eisenhauer [iD] http://orcid.org/0000-0002-0371-6720

### Decision letter and Author response

Decision letter https://doi.org/10.7554/eLife.55659.sa1
Author response https://doi.org/10.7554/eLife.55659.sa2

## Additional files

### Supplementary files

• Transparent reporting form

### Data availability

Data and codes for the analyses are available on the iDiv Data repository (DOI: https://doi.org/10.25829/idiv.1868-15-3033) and GitHub (https://github.com/leabeaumelle/BEFunderGlobalChange; copy archived at https://github.com/elifesciences-publications/BEFunderGlobalChange).

The following dataset was generated:

| Author(s) | Year | Dataset title | Dataset URL | Database and Identifier |
|---|---|---|---|---|
| Beaumelle L, De Laender F, Eisenhauer N | 2020 | Biodiversity mediated effects of stressors and nutrients on decomposition | https://doi.org/10.25829/idiv.1868-15-3033 | German Centre for Integrative Biodiversity Research (iDiv) Halle-Jena-Leipzig, 10.25829/idiv.1868-15-3033 |

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

## Appendix 1

## Supplementary Methods

### Search strategy

We collected data from published papers reporting the effect of various global change drivers on both decomposition rates and decomposer communities. The search strategy first involved the selection of a relevant search term combination. We compared different search term combinations based on the number of studies retrieved, their potential relevance (based on screening the titles in the search), and on maximizing the retrieval of pre-identified papers that fully matched the inclusion criteria. We used search terms of previous meta-analyses and literature reviews (*Covich et al., 2004*; *García-Palacios et al., 2015*; *Srivastava et al., 2009*). The following search terms were used to identify studies looking at the impact of various global change drivers on both decomposition rates and decomposer communities. The search was done on ISI Web Of Science on November 17th 2017 and retrieved 2536 records.

TS=("global change" OR "environmental change" OR disturbance* OR stress* OR "climat* change" OR drought OR temperature* OR warming OR heat* OR precipitation* OR rain* OR flood* OR irrigation OR moisture OR watering OR fire OR "carbon dioxide" OR CO2 OR acidification OR "nitrogen deposition" OR "nutrient deposition" OR "atmospheric deposition" OR *eutroph* OR fertili* OR "nutrient* enrichment" OR "nutrient pollut*" OR "land-use" OR "landuse" OR "agricultural intensi*" OR desertif* OR pollut* OR pesticide* OR metal* OR "over-exploit*" OR overexploit* OR toxi* OR contamin* OR over-fish* OR invasi* OR alien OR "habitat loss" OR "habitat fragment*" OR "habitat degrad*" OR "habitat destruct*" )

AND

TS = ((decomposition OR processing OR breakdown OR decay OR 'mass loss') AND (litter OR leaf OR leaves OR bark OR wood))

AND

TS = (("species richness" OR richness OR "number of species" OR "number of taxa" OR "species diversity" OR "taxonomic diversity" OR biodiversity OR Shannon* OR even- ness OR "community composition" OR "community structure" OR "functional diversity" OR "trait diversity" OR "functional traits" OR "functional group richness" OR "trait-based") AND (decomposer* OR detritivore* OR *invertebrate* OR microb* OR microorganism* OR bacteri* OR fung* OR archaea OR shredder OR *invertebrate* OR hyphomycete* OR "leaf-shredding" OR "leaf-eat*" OR "leaf-consum*" "leaf-feed*" OR "litter-feed*" OR "litter-eat*" OR "litter-shredding" OR protozoa* OR protist* OR springtail OR collembol* OR mite* OR acari* OR enchytraeid* OR nematod* OR rotifer* OR isopod* OR earthworm* OR termite* OR microarthropod* OR macroarthropod* OR microfauna OR mesofauna OR macrofauna)).

Abstracts were individually screened using the online software Abstrackr (http://abstrackr. cebm.brown.edu/account/login) to identify references matching our inclusion criteria. At the screening step, tags were given to classify studies according to the type of drivers. This step resulted in 384 articles potentially relevant for the meta-analysis, 2152 abstracts did not match the inclusion criteria (mostly because they were not looking at both decomposition rates and decomposer communities responses to global change, or because they manipulated decomposer communities directly).

*Appendix 1—figure 1* reports the PRISMA diagram describing the different steps to assemble our datasets. After full text screening of the 112 potentially relevant papers, 61 papers verified our inclusion criteria and reported data that we could extract for the meta-analysis. For the SEM analysis, two papers were further excluded because some data needed for the models were missing (typically the levels of nutrients or stressors).

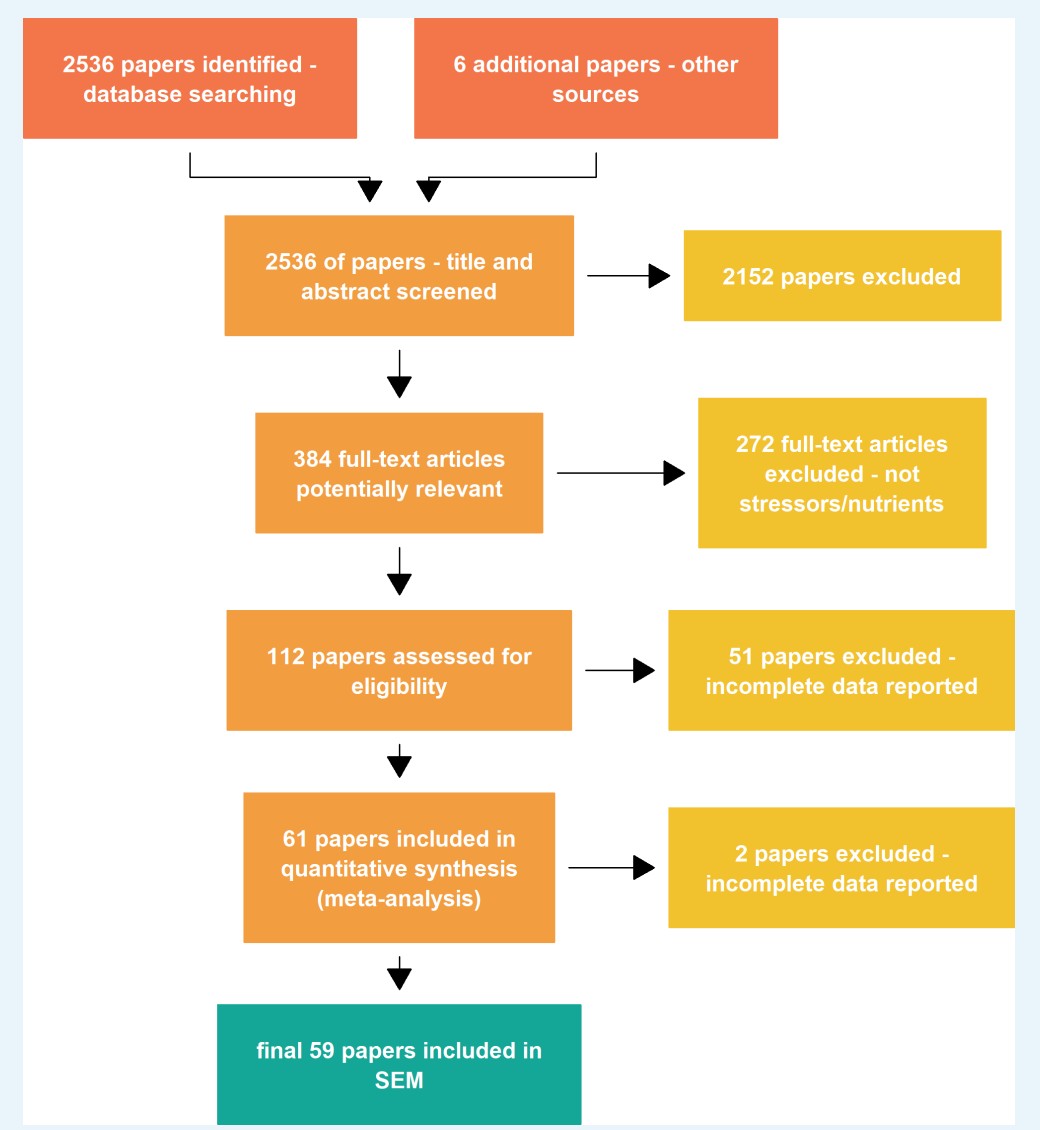

**Appendix 1—figure 1.** PRISMA plot describing the data collection steps of the meta-analysis. SEM = structural equation modeling.

## Environmental quality standards

**Appendix 1—table 1. Environmental quality criteria for stressors and nutrients.** Quality criteria were used to standardized the intensity levels of the different chemical stressors across studies included in the meta-analysis.

| System | Chemical or nutrient | Unit1 | Unit2 | Quality criteria | Citation |
|---|---|---|---|---|---|
| aquatic | fungicide: pyrimethanil | μg/l | - | 0.69 | Abelho M, Martins TF, Shinn C, Moreira-Santos M, Ribeiro R. 2016. Effects of the fungicide pyrimethanil on biofilm and organic matter processing in outdoor lentic mesocosms. Ecotoxicology 25:121–131. |
| aquatic | fungicide: tebuconazole | μg/l | - | 0.10 | https://echa.europa.eu/documents/10162/41e9d7aa-4559-f904-9cb5-0a0d5f0d6445 |

*Appendix 1—table 1 continued on next page*

*Appendix 1—table 1 continued*

| System | Chemical or nutrient | Unit1 | Unit2 | Quality criteria | Citation |
|---|---|---|---|---|---|
| aquatic | As | µg/l | - | 13.00 | https://echa.europa.eu/brief-profile/-/brief-profile/100.028.316 |
| aquatic | Al | µg/l | - | 87.00 | https://www.govinfo.gov/content/pkg/FR-2018-12-21/pdf/2018-27745.pdf |
| aquatic | Cu | µg/l | - | 10.10 | https://echa.europa.eu/brief-profile/-/brief-profile/100.124.825 |
| aquatic | Zn | µg/l | - | 20.60 | https://echa.europa.eu/brief-profile/-/brief-profile/100.028.341 |
| aquatic | Fe | µg/l | - | 1000.00 | https://www.epa.gov/wqc/national-recommended-water-quality-criteria-aquatic-life-criteria-table |
| aquatic | Mn | µg/l | - | 1000.00 | https://www.epa.gov/wqc/national-recommended-water-quality-criteria-aquatic-life-criteria-table |
| aquatic | Hg | µg/l | - | 0.06 | https://echa.europa.eu/brief-profile/-/brief-profile/100.028.278 |
| aquatic | Cd | µg/l | - | 0.19 | https://echa.europa.eu/brief-profile/-/brief-profile/100.028.320 |
| aquatic | insecticide: chlorpyrifos | µg/l | - | 0.08 | https://www.epa.gov/wqc/national-recommended-water-quality-criteria-aquatic-life-criteria-table |
| aquatic | phenanthrene | µg/l | - | 51.40 | Wu, J. Y., Yan, Z. G., Liu, Z. T., Liu, J. D., Liang, F., Wang, X. N., & Wang, W. L. (2015). Development of water quality criteria for phenanthrene and comparison of the sensitivity between native and non-native species. *Environmental Pollution*, 196, 141-146. |
| aquatic | Zn | mg/kg | - | 117.80 | https://echa.europa.eu/brief-profile/-/brief-profile/100.028.341 |
| aquatic | Cd | mg/kg | - | 1.80 | https://echa.europa.eu/brief-profile/-/brief-profile/100.028.320 |
| aquatic | Hg | mg/kg | - | 9.30 | https://echa.europa.eu/brief-profile/-/brief-profile/100.028.278 |
| aquatic | Pb | mg/kg | - | 186.00 | https://echa.europa.eu/brief-profile/-/brief-profile/100.028.273 |
| terrestrial | Cu | mg/kg | - | 106.35 | https://echa.europa.eu/brief-profile/-/brief-profile/100.124.825 |
| terrestrial | Zn | mg/kg | - | 35.60 | https://echa.europa.eu/brief-profile/-/brief-profile/100.028.341 |
| terrestrial | Ni | mg/kg | - | 29.90 | https://echa.europa.eu/brief-profile/-/brief-profile/100.028.283 |
| terrestrial | Mn | mg/kg | - | 3.40 | https://echa.europa.eu/brief-profile/-/brief-profile/100.028.277 |
| terrestrial | Hg | µg/kg | - | 22.00 | https://echa.europa.eu/brief-profile/-/brief-profile/100.028.278 |
| terrestrial | Pb | mg/kg | - | 212.00 | https://echa.europa.eu/brief-profile/-/brief-profile/100.028.273 |
| terrestrial | Cd | mg/kg | - | 0.90 | https://echa.europa.eu/brief-profile/-/brief-profile/100.028.320 |
| terrestrial | insecticide: chlorpyrifos | kg/ha | - | 1.25 | Iwai CB, Noller B. 2010. Ecotoxicological assessment of diffuse pollution using bio-monitoring tool for sustainable land use in Thailand. Journal of Environmental Sciences 22:858–863. |

Appendix 1—table 1 continued

| System | Chemical or nutrient | Unit1 | Unit2 | Quality criteria | Citation |
|---|---|---|---|---|---|
| terrestrial | insecticide: endosulfan | kg/ha | - | 1.25 | Iwai CB, Noller B. 2010. Ecotoxicological assessment of diffuse pollution using bio-monitoring tool for sustainable land use in Thailand. Journal of Environmental Sciences 22:858–863. |
| terrestrial | herbicide: atrazine | kg/ha | - | 1.88 | Iwai CB, Noller B. 2010. Ecotoxicological assessment of diffuse pollution using bio-monitoring tool for sustainable land use in Thailand. Journal of Environmental Sciences 22:858–863. |
| terrestrial | insecticide: carbofuran | kg/ha | - | 31.25 | Iwai CB, Noller B. 2010. Ecotoxicological assessment of diffuse pollution using bio-monitoring tool for sustainable land use in Thailand. Journal of Environmental Sciences 22:858–863. |
| aquatic | pesticide mixture | arbitrary | - | 1.00 | Talk A. 2016. Effects of multiple butLow pesticide loads on aquatic fungal communities colonizing leaf litter. Journal of EnvironmentalSciences 46:116–125. |
| terrestrial | herbicide: glyphosate | kg/ha | - | 4.32 | European Food Safety Authority (EFSA). Conclusion on the peer review of the pesticide risk assessment of the active substance glyphosate. EFSA Journal 13, (2015). |
| terrestrial | herbicide: simazine | kg/ha | - | 0.10 | https://ec.europa.eu/food/plant/pesticides/eu-pesticides-database/public/?event = activesubstance.detail and language = EN and selectedID = 1853 |
| aquatic | pesticide mixture | sum or max of TU (toxic units) | - | −3.50 | Schäfer, R. B., Caquet, T., Siimes, K., Mueller, R., Lagadic, L., & Liess, M. (2007). Effects of pesticides on community structure and ecosystem functions in agricultural streams of three biogeographical regions in Europe. *Science of the Total Environment*, *382*(2-3), 272-285. |
| aquatic | DIN | mg/l | N | 3.05 | Ministère de l'Environnement, de l'Énergie et de la Mer. Guide technique Relatif à l'évaluation de l'état des eaux de surface continen- tales (cours d'eau, canaux, plans d'eau). (2016). |
| aquatic | NH4+ | mg/l | $NH_4$ | 0.10 | Ministère de l'Environnement, de l'Énergie et de la Mer. Guide technique Relatif à l'évaluation de l'état des eaux de surface continen- tales (cours d'eau, canaux, plans d'eau). (2016). |
| aquatic | NO3 | mg/l | $NO_3$ | 10.00 | Ministère de l'Environnement, de l'Énergie et de la Mer. Guide technique Relatif à l'évaluation de l'état des eaux de surface continen- tales (cours d'eau, canaux, plans d'eau). (2016). |
| aquatic | NO2 | mg/l | $NO_2$ | 0.10 | Ministère de l'Environnement, de l'Énergie et de la Mer. Guide technique Relatif à l'évaluation de l'état des eaux de surface continen- tales (cours d'eau, canaux, plans d'eau). (2016). |
| aquatic | Total_N | mg/l | N | 0.67 | US EPA, O. Water Quality Criteria. US EPA (2013). Available at: https://www.epa.gov/wqc. (Accessed: 7th January 2019) |

*Appendix 1—table 1 continued*

| System | Chemical or nutrient | Unit1 | Unit2 | Quality criteria | Citation |
|---|---|---|---|---|---|
| aquatic | SRP | mg/l | $PO_4^3$ | 0.10 | Guide technique Relatif à l'évaluation de l'état des eaux de surface continen- tales (cours d'eau, canaux, plans d'eau). (Minis- tère de l'Environnement, de l'Énergie et de la Mer, 2016). |
| aquatic | Total_P | mg/l | P | 0.05 | Guide technique Relatif à l'évaluation de l'état des eaux de surface continen- tales (cours d'eau, canaux, plans d'eau). (Minis- tère de l'Environnement, de l'Énergie et de la Mer, 2016). |
| terrestrial | N deposition | kg/ha/yr | N | 20.00 | Pardo, L.H., Fenn, M.E., Goodale, C.L., Geiser, L.H., Driscoll, C.T., Allen, E.B., Baron, J.S., Bobbink, R., Bowman, W.D., Clark, C.M., Emmett, B., Gilliam, F.S., Greaver, T.L., Hall, S.J., Lilleskov, E.A., Liu, L., Lynch, J.A., Nadelhoffer, K.J., Perakis, S. S., Robin-Abbott, M.J., Stoddard, J.L., Weathers, K.C. and Dennis, R.L. (2011), Effects of nitrogen deposition and empirical nitrogen critical loads for ecoregions of the United States. Ecological Applications, 21: 3049-3082. doi:10.1890/10-2341.1; derived critical loads (i.e. level of deposition below which no detrimental ecological effect occurs over the long term according to current knowledge) from empirical data for various (plant) species and ecosystems |
| terrestrial | P fertilization | kg/ha/yr | P | 35.00 | Amery, F., & Schoumans, O. F. (2014). *Agricultural phosphorus legislation in Eur- ope.* Institute for Agricultural and Fisheries Research (ILVO). |

## Appendix 2

### Meta-analysis

### Publication bias

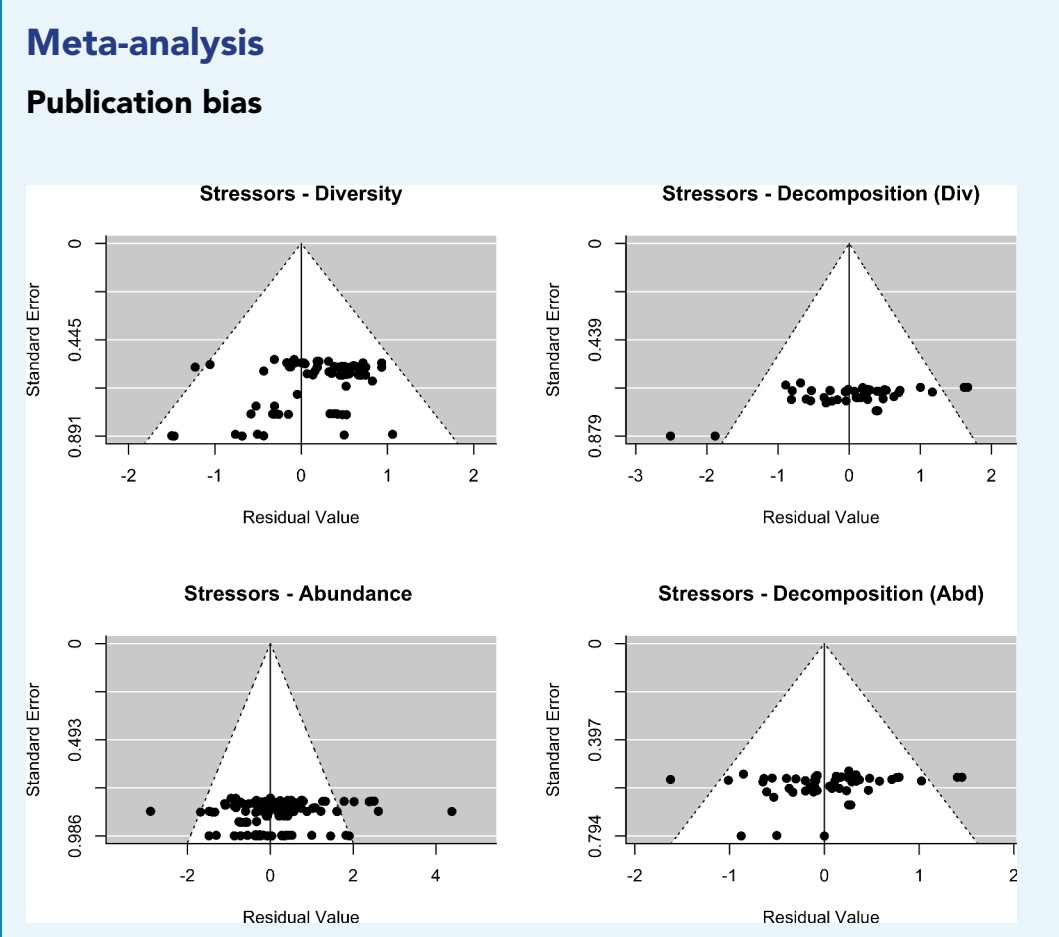

**Appendix 2—figure 1.** Assessment of publication bias. Stressors: Funnel plots of each response variables (decomposer diversity, abundance and decomposition) in the two datasets (stressors - diversity and stressors - abundance). Meta-analytic models included the effect of stressor intensity (standardized levels) as a covariate.

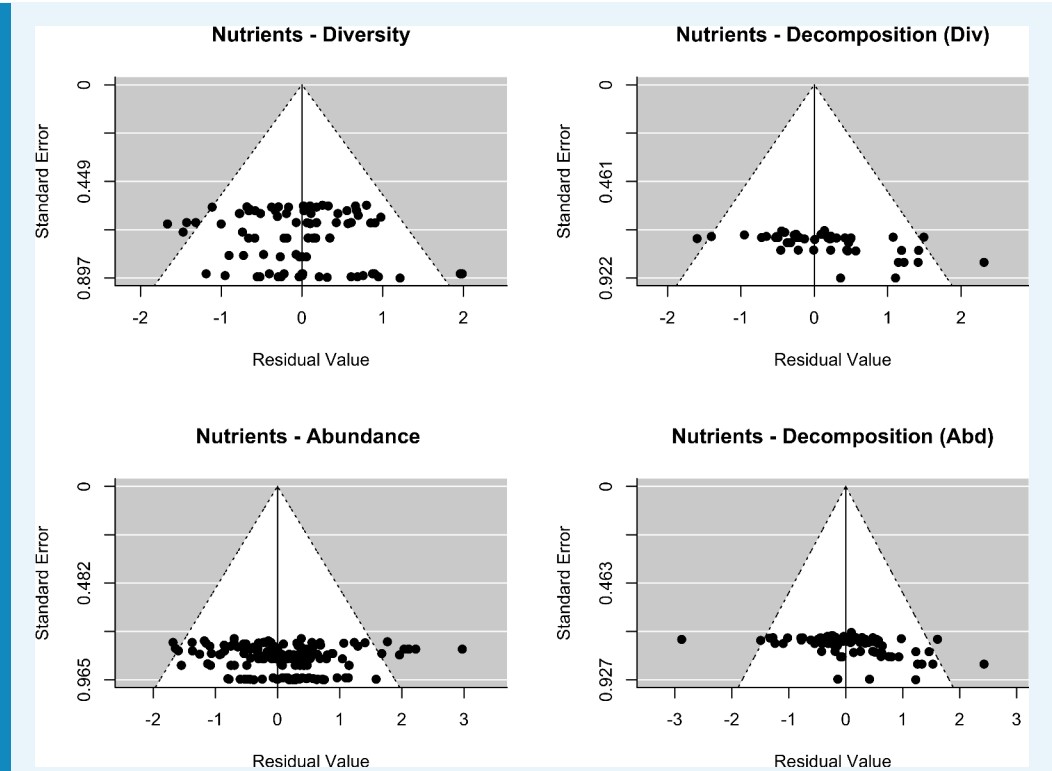

**Appendix 2—figure 2.** Assessment of publication bias. Nutrients: Funnel plots of each response variables (decomposer diversity, abundance and decomposition) in the two datasets (stressors - diversity and stressors - abundance). Meta-analytic models included the effect of nutrient intensity (standardized levels) as a covariate.

**Appendix 2—table 1.** Assessment of publication bias. Results from Egger's regressions showing the intercepts, standard error (SE) and p-value of regressions between standard normal deviate of each response variable (effect sizes) and the inverse of their standard errors. Models also included stressor or nutrient intensity as a covariate.

| Dataset | Variable | Publication bias p | Publication bias | Intercept | SE |
|---|---|---|---|---|---|
| Stressors - Biodiv | Biodiversity | 0.10 | no | −1.36 | 0.83 |
| Stressors - Biodiv | Decomposition | 0.58 | no | −1.07 | 1.94 |
| Stressors - Abdc | Abundance | 0.14 | no | −1.49 | 1.02 |
| Stressors - Abdc | Decomposition | 0.68 | no | −0.67 | 1.60 |
| Nutrients - Biodiv | Biodiversity | 0.37 | no | 0.76 | 0.86 |
| Nutrients - Biodiv | Decomposition | 0.19 | no | 3.35 | 2.55 |
| Nutrients - Abdc | Abundance | 0.08 | no | 1.21 | 0.70 |
| Nutrients - Abdc | Decomposition | <0.001 | pub. bias | 5.31 | 1.45 |

## Meta-analysis - first level: overall mean effects

**Appendix 2—table 2. First level meta-analysis comparing the effects of chemical stressors and nutrient enrichment.** Results of Wald-type chi-square tests comparing the grand mean effect sizes of the three response variables (decomposer diversity, abundance and litter decomposition) between chemical stressors and nutrient enrichment.

| Response | QM | Df | N | p-value |
|---|---|---|---|---|
| Diversity | 25.65 | 2 | 174 | <0.001 |
| Abundance | 7.92 | 2 | 424 | 0.019 |
| Litter decomposition | 17.61 | 2 | 165 | <0.001 |

## SEM analysis

**Appendix 2—table 3. Summary table of structural equation modelling (SEM) analysis.** Unstandardized path coefficients from SEMs for the four datasets: Stressors - Biodiversity (Biodiv), Stressors - Abundance (Abdc), Nutrients - Biodiversity and Nutrients, Abundance. SEMs also incorporated categorical predictors (study type, taxonomic group and diversity metric, see Materials and methods).

| Dataset | Response | Predictor | Estimate | SE | Crit.value | Df | p-Value |
|---|---|---|---|---|---|---|---|
| Stressors - Biodiv | Decomposition | Diversity | 0.42 | 0.17 | 2.50 | 19 | 0.022 |
| Stressors - Biodiv | Decomposition | Stressor intensity | −0.02 | 0.04 | −0.47 | 19 | 0.643 |
| Stressors - Biodiv | Diversity | Stressor intensity | −0.10 | 0.04 | −2.44 | 18 | 0.025 |
| Stressors - Abdc | Decomposition | Abundance | 0.24 | 0.08 | 2.97 | 25 | 0.007 |
| Stressors - Abdc | Decomposition | Stressor intensity | −0.01 | 0.03 | −0.41 | 25 | 0.683 |
| Stressors - Abdc | Abundance | Stressor intensity | 0.00 | 0.05 | 0.03 | 25 | 0.977 |
| Nutrients - Biodiv | Decomposition | Diversity | 0.01 | 0.11 | 0.06 | 20 | 0.951 |
| Nutrients - Biodiv | Decomposition | Nutrient intensity | −0.08 | 0.06 | −1.21 | 20 | 0.239 |
| Nutrients - Biodiv | Diversity | Nutrient intensity | −0.25 | 0.07 | −3.51 | 19 | 0.002 |
| Nutrients - Abdc | Decomposition | Abundance | 0.08 | 0.10 | 0.76 | 44 | 0.451 |
| Nutrients - Abdc | Decomposition | Nutrient intensity | −0.12 | 0.05 | −2.16 | 44 | 0.037 |
| Nutrients - Abdc | Abundance | Nutrient intensity | −0.06 | 0.06 | −1.00 | 44 | 0.321 |

## Meta-analysis - second-level: categorical moderators

**Appendix 2—table 4. Main effects of categorical predictors on decomposer diversity, abundance and decomposition in the four datasets: Stressors - Biodiversity (Biodiv), Stressors - Abundance (Abdc), Nutrients - Biodiversity and Nutrients, Abundance.** Results are QM statistics and associated p-values of the second-level meta-analyses.

| Dataset | Response | Predictor | QM | p-value |
|---|---|---|---|---|
| Stressors - Biodiv | Diversity | Taxonomic group | 4.80 | 0.028 |
| Stressors - Abdc | Abundance | Taxonomic group | 10.10 | 0.001 |
| Nutrients - Biodiv | Diversity | Taxonomic group | 12.77 | <0.001 |
| Nutrients - Abdc | Abundance | Taxonomic group | 4.53 | 0.033 |
| Stressors - Biodiv | Diversity | Study type | 1.89 | 0.169 |
| Stressors - Abdc | Abundance | Study type | 0.92 | 0.338 |
| Nutrients - Biodiv | Diversity | Study type | 0.24 | 0.625 |
| Nutrients - Abdc | Abundance | Study type | 0.98 | 0.323 |
| Stressors - Biodiv | Diversity | Diversity metric | 1.67 | 0.196 |
| Nutrients - Biodiv | Diversity | Diversity metric | 2.35 | 0.125 |
| Stressors - Biodiv | Decomposition | Study type | 0.16 | 0.693 |
| Stressors - Abdc | Decomposition | Study type | 1.85 | 0.174 |
| Nutrients - Biodiv | Decomposition | Study type | 2.69 | 0.101 |
| Nutrients - Abdc | Decomposition | Study type | 0.18 | 0.674 |

## Appendix 3

### Sensitivity analyses

### Influence of approximating standard deviations

When studies did not report standard deviations associated with the mean decomposer diversity or abundance or the mean decomposition rates, we used linear approximations to estimate the variance based on our data (see Materials and methods). We tested the influence of those approximations on the final results by running the structural equation modelling (SEM) analysis without those effect sizes for which standard deviations were approximated. Overall, the same patterns were found showing that approximating missing standard deviations had limited effects on the final SEM results.

**Appendix 3—table 1.** Results of mediation tests from structural equation modeling (SEM) analysis based on data without approximated standard deviations. C statistic and associated p-value for SEM without the path from biodiversity or abundance to decomposition for the four datasets: Stressors - Diversity, Stressors - Abundance, Nutrients - Diversity and Nutrients - Abundance. ΔAIC is the difference in AIC score between models with and without biodiversity- or abundance-mediated effects.

| Dataset | C statistic | Df | p-value | ΔAIC | No. of studies | N |
|---|---|---|---|---|---|---|
| Stressors, Biodiv | 12.42 | 6 | 0.053 | −8.32 | 16 | 58 |
| Stressors, Abdc | 10.15 | 4 | 0.038 | −6.82 | 23 | 216 |
| Nutrient, Biodiv | 13.33 | 6 | 0.038 | −1.46 | 21 | 67 |
| Nutrient, Abdc | 3.82 | 4 | 0.432 | −0.12 | 32 | 127 |

**Appendix 3—table 2.** Summary table of structural equation modeling (SEM) analysis based on data without approximated standard deviations. Standardized (Std.est.) and unstandardized estimate (Est.) path coefficients from SEMs for the four datasets.

| Dataset | Response | Predictor | Std. est. | Est. | SE | Crit. value | Df | p-value |
|---|---|---|---|---|---|---|---|---|
| Stress., Biodiv | Decomposition | Diversity | 0.52 | 0.50 | 0.16 | 3.16 | 12 | 0.008 |
| Stres., Biodiv | Decomposition | Stressor intensity | −0.26 | −0.05 | 0.03 | −1.54 | 12 | 0.148 |
| Stress., Biodiv | Diversity | Stressor intensity | −0.39 | −0.08 | 0.04 | −1.89 | 11 | 0.085 |
| Stress., Abdc | Decomposition | Abundance | 0.40 | 0.27 | 0.09 | 2.91 | 19 | 0.009 |
| Stress., Abdc | Decomposition | Stressor intensity | −0.11 | −0.02 | 0.03 | −0.77 | 19 | 0.450 |
| Stress., Abdc | Abundance | Stressor intensity | 0.08 | 0.03 | 0.06 | 0.46 | 19 | 0.649 |
| Nut., Biodiv | Decomposition | Diversity | −0.04 | −0.04 | 0.12 | −0.35 | 10 | 0.732 |
| Nut., Biodiv | Decomposition | Nutrient intensity | −0.31 | −0.14 | 0.09 | −1.52 | 10 | 0.161 |
| Nut., Biodiv | Diversity | Nutrient intensity | −0.49 | −0.23 | 0.10 | −2.39 | 9 | 0.040 |
| Nut., Abdc | Decomposition | Abundance | 0.05 | 0.04 | 0.13 | 0.33 | 29 | 0.742 |
| Nut., Abdc | Decomposition | Nutrient intensity | −0.26 | −0.12 | 0.06 | −1.91 | 29 | 0.066 |
| Nut., Abdc | Abundance | Nutrient intensity | −0.20 | −0.10 | 0.07 | −1.40 | 29 | 0.173 |

### Influence of extreme values

We re-run our SEMs with datasets excluding extreme values of effect sizes. Extreme values were defined as values exceeding the whiskers of boxplots. Overall, we found similar patterns showing that extreme effect sizes had limited effects on the final SEM results.

**Appendix 3—table 3.** Results of mediation tests from structural equation modeling (SEM) analysis based on data excluding extreme values of effect sizes. C statistic and associated p-value for SEM without the path from biodiversity or abundance to decomposition for the four datasets: Stressors - Diversity, Stressors - Abundance, Nutrients - Diversity and Nutrients - Abundance. ΔAIC is the difference in AIC score between models with and without biodiversity- or abundance-mediated effects.

| Dataset | C statistic | Df | p-value | ΔAIC | No. of studies | N |
|---|---|---|---|---|---|---|
| Stressors, Biodiv | 10.18 | 6 | 0.117 | −6.71 | 22 | 94 |
| Stressors, Abdc | 7.39 | 4 | 0.117 | −4.23 | 27 | 254 |
| Nutrient, Biodiv | 14.80 | 6 | 0.022 | −4.85 | 26 | 93 |
| Nutrient, Abdc | 2.74 | 4 | 0.603 | 0.15 | 35 | 159 |

**Appendix 3—table 4.** Summary table of structural equation modelling (SEM) analysis based on data excluding extreme values of effect sizes. Standardized (Std.est.) and unstandardized estimate (Est.) path coefficients from SEMs for the four datasets.

| Dataset | Response | Predictor | Std. est. | Est. | SE | Crit. value | Df | p-value |
|---|---|---|---|---|---|---|---|---|
| Stress., Biodiv | Decomposition | Diversity | 0.41 | 0.40 | 0.18 | 2.20 | 18 | 0.041 |
| Stress., Biodiv | Decomposition | Stressor intensity | −0.04 | −0.01 | 0.04 | −0.24 | 18 | 0.814 |
| Stress., Biodiv | Diversity | Stressor intensity | −0.44 | −0.10 | 0.04 | −2.75 | 17 | 0.014 |
| Stress., Abdc | Decomposition | Abundance | 0.30 | 0.24 | 0.11 | 2.24 | 23 | 0.035 |
| Stress., Abdc | Decomposition | Stressor intensity | 0.05 | 0.01 | 0.03 | 0.35 | 23 | 0.731 |
| Stress., Abdc | Abundance | Stressor intensity | 0.00 | 0.00 | 0.04 | −0.02 | 23 | 0.980 |
| Nut., Biodiv | Decomposition | Diversity | 0.00 | 0.00 | 0.11 | 0.02 | 19 | 0.986 |
| Nut., Biodiv | Decomposition | Nutrient intensity | −0.18 | −0.08 | 0.06 | −1.30 | 19 | 0.210 |
| Nut., Biodiv | Diversity | Nutrient intensity | −0.53 | −0.24 | 0.07 | −3.36 | 18 | 0.003 |
| Nut., Abdc | Decomposition | Abundance | 0.00 | 0.00 | 0.09 | 0.04 | 37 | 0.968 |
| Nut., Abdc | Decomposition | Nutrient intensity | −0.38 | −0.13 | 0.04 | −3.26 | 37 | 0.002 |
| Nut., Abdc | Abundance | Nutrient intensity | −0.24 | −0.09 | 0.05 | −1.73 | 37 | 0.092 |

## Influence of the effect size metric

We tested the influence of the metric of effect size selected on the results of the SEMs. Log-response ratios were calculated instead of correlation coefficients and the models were re-run based on those data. The results were partially different from the original analysis. For nutrients, similar patterns were found, however for stressors there was limited support for the biodiversity- and abundance-mediated effects on decomposition responses. We noted extreme values of log-response ratios that may have explained such patterns. Besides, the log-response ratio has a different interpretation compared to correlation coefficients. Log-response ratios are sensitive to the different metrics of diversity and abundance, taxa groups, litter types etc. used across studies included in this meta-analysis. Therefore, this result reinforced our choice of correlation coefficients as relevant effect sizes in the present meta-analysis.

**Appendix 3—table 5.** Results of mediation tests from structural equation modeling (SEM) analysis based on data using log-response ratio as an effect size. C statistic and associated p-value for SEM without the path from biodiversity or abundance to decomposition for the four datasets: Stressors - Diversity, Stressors - Abundance, Nutrients - Diversity and Nutrients - Abundance. ΔAIC is the difference in AIC score between models with and without biodiversity- or abundance-mediated effects.

| Dataset | C statistic | Df | p-value | ΔAIC | No. of studies | N |
|---|---|---|---|---|---|---|
| Stressors, Biodiv | 4.11 | 6 | 0.662 | −0.02 | 22 | 70 |
| Stressors, Abdc | 5.59 | 4 | 0.232 | −2.22 | 37 | 150 |
| Nutrient, Biodiv | 8.03 | 6 | 0.236 | −2.08 | 14 | 78 |
| Nutrient, Abdc | 3.41 | 4 | 0.492 | −0.44 | 21 | 307 |

**Appendix 3—table 6.** Summary table of structural equation modeling (SEM) analysis based on data using log-response ratio as an effect size. Standardized (Std.est.) and unstandardized estimate (Est.) path coefficients from SEMs for the four datasets.

| Dataset | Response | Predictor | Std.est | Est. | SE | Crit.value | Df | p-value |
|---|---|---|---|---|---|---|---|---|
| Stress., Biodiv | Decomposition | Diversity | 0.18 | 0.12 | 0.15 | 0.80 | 15 | 0.437 |
| Stress., Biodiv | Decomposition | Stressor intensity | −0.24 | −0.05 | 0.04 | −1.47 | 15 | 0.163 |
| Stress., Biodiv | Diversity | Stressor intensity | −0.35 | −0.12 | 0.03 | −4.17 | 15 | 0.001 |
| Stress., Abdc | Decomposition | Abundance | 0.14 | 0.04 | 0.05 | 0.86 | 28 | 0.396 |
| Stress., Abdc | Decomposition | Stressor intensity | 0.09 | 0.02 | 0.04 | 0.55 | 28 | 0.586 |
| Stress., Abdc | Abundance | Stressor intensity | −0.14 | −0.11 | 0.11 | −1.03 | 28 | 0.312 |
| Nut., Biodiv | Decomposition | Diversity | 0.29 | 0.19 | 0.10 | 1.80 | 14 | 0.094 |
| Nut., Biodiv | Decomposition | Nutrient intensity | −0.15 | −0.07 | 0.08 | −0.96 | 14 | 0.352 |
| Nut., Biodiv | Diversity | Nutrient intensity | −0.20 | −0.16 | 0.07 | −2.11 | 14 | 0.054 |
| Nut., Abdc | Decomposition | Abundance | 0.06 | 0.04 | 0.06 | 0.59 | 42 | 0.559 |
| Nut., Abdc | Decomposition | Nutrient intensity | −0.36 | −0.16 | 0.05 | −3.08 | 42 | 0.004 |
| Nut., Abdc | Abundance | Nutrient intensity | −0.01 | 0.00 | 0.08 | −0.08 | 42 | 0.935 |

## Appendix 4

### References included in the meta-analysis

*Abelho et al., 2016*
*Allison et al., 2010*
*Artigas et al., 2012*;
*Bergfur et al., 2007*
*Bott et al., 2012*;
*Brosed et al., 2016*
*Cabrini et al., 2013*;
*Carlisle and Clements, 2005*;
*Chaffin et al., 2005*;
*Chung and Suberkropp, 2008*;
*Creamer et al., 2008*;
*Duarte et al., 2004*;
*Duarte et al., 2009b*;
*Duarte et al., 2008*;
*Duarte et al., 2009a*;
*Dunck et al., 2015*;
*Englert et al., 2013*;
*Estevez et al., 2017*;
*Fanin, 2015*;
*Fernandes et al., 2009*;
*Ferreira et al., 2006*;
*Fernández et al., 2016*
*Gan et al., 2013*;
*Gulis et al., 2006*;
*Gulis and Suberkropp, 2003*;
*Hagen et al., 2006*;
*Hobbie et al., 2012*;
*Hobbelen et al., 2006*
*Hogsden and Harding, 2013*;
*Hopkins et al., 2011*;
*Iwai and Noller, 2010*;
*Johnson et al., 2014*;
*Lecerf et al., 2006*;
*Lima-Fernandes et al., 2015*;
*Liu et al., 2010*;
*Lopes et al., 2017*;
*Lucisine et al., 2015*
*Matthaei et al., 2010*;
*Mederiros, 2015*;
*Menéndez, 2009*;
*Mesquita et al., 2007*;
*Milcu et al., 2011*;
*Moreirinha et al., 2011*;
*Münze et al., 2015*;
*Niyogi et al., 2001*;
*Niyogi et al., 2009*
*Ouédraogo et al., 2004*;
*Pascoal et al., 2003*;
*Pearson and Connolly, 2000*;
*Pérez et al., 2013*;
*Perkiömäki et al., 2003*;
*Piscart et al., 2009*;
*Quintão et al., 2013*;
*Schultheis, 1997*;
*Shaftel et al., 2011*;

*Schäfer et al., 2007*;
*Talk, 2016*;
*Thompson, 2016*;
*Wardle et al., 2001*;
*Woodcock and Huryn, 2005*;
*Woodward et al., 2012*

