## [Decision Letter]

**Acceptance summary:**

Many ecosystems are severely affected by human activity, yet we still know surprisingly little about the modulating role of specific elements on ecosystem health on a global scale. This paper provides a convincing and timely synthesis of the effects of chemical stressors and nutrient enrichment on soil biodiversity, and a major ecosystem process they modulate – litter decomposition. This work will inspire new research linking ecological stoichiometry to ecosystem services.

**Decision letter after peer review:**

Thank you for submitting your article "Biodiversity mediates the effects of stressors but not nutrients on litter decomposition" for consideration by *eLife*. Your article has been reviewed by two peer reviewers, and the evaluation has been overseen by a Reviewing Editor and Christian Rutz as the Senior Editor. The reviewers have opted to remain anonymous.

The reviewers have discussed their reports with one another, and the Reviewing Editor has drafted this decision letter to help you prepare a revised submission.

Summary:

The paper describes a meta-analysis on the effects of nutrients and chemical stressors on biodiversity and the relationship with ecosystem functioning. Using two meta-analyses and Structural Equation Modelling, they find that chemical stressors (but not added nutrients) affect leaf litter decomposition, through changes in biodiversity and abundance of organisms. Interestingly, the authors do not establish a stressor intensity-effect relationship. They find minor effects of nutrients on the biodiversity-ecosystem functioning relationship, though they establish relationships with nutrient intensity, depicting real-world scenarios of human impacts on ecosystems.

Essential revisions:

1) The authors run different analyses – for the average reader it may be difficult to understand why there is a stressor effect in the SEM, but no stressor intensity-effect relationship. We find this aspect under-appreciated in the Discussion, and it could be described more clearly what the reasons for this difference are. In addition, it should be discussed to which extent the variable basis for the values used to characterize stressor intensity may be the reason for this finding. There is a relationship for nutrients, where the authors have a relatively consistent data source, whereas, for chemical stressors where the data sources vary, including the methods used to derive quality criteria, they found no relationship. Our suspicion is therefore that this is partly due to methodical reasons. To clarify, problems may arise if you choose a benchmark Y for stressor S1, but multiple benchmarks Zi where i from 1 to n for Stressor S2i that are not consistent with each other. Now imagine you find a relationship for one but not for the other with stressor intensity – this can simply be because you have variable benchmarks Zi and is not necessarily due to the stressor.

2) The stressor definition should be reconsidered. The authors define: "stressors (e.g., temperature, drought, chemicals) and resource enrichment (e.g., of CO_2_ or mineral nutrients)." From a theoretical perspective, the temperature would also be a niche-defining resource. From a practical perspective, it is confusing for the reader to contrast "chemical stressors" to CO_2_ and NO_3_, the latter of which are clearly chemicals and often anthropogenic. Then both are "environmental change drivers", which makes the terminology even more complicated for the reader. Why not just use stressors and environmental change drivers, which have different niche-defining properties? Some are unimodal (nutrients, temperature), others exhibit a threshold relationship and others may be log-linear.

---

## [Author Response]

Essential revisions:1) The authors run different analyses – for the average reader it may be difficult to understand why there is a stressor effect in the SEM, but no stressor intensity-effect relationship. We find this aspect under-appreciated in the Discussion, and it could be described more clearly what the reasons for this difference are.

We agree that the Discussion was not clear enough regarding this important point. We conducted two approaches for analyzing the data: an SEM and a meta-regression. The results are slightly different because different datasets were used. For the SEM, we used data resampling. We tested 1000 iterations of the model, each iteration based on a random sample of the effect sizes on biodiversity or abundance. This approach was done to account for the fact that studies often reported multiple measures of biodiversity or abundance response in the same litterbag where litter decomposition was assessed. Thus, the data had many cases where a unique effect size on decomposition was associated with multiple effect sizes on biodiversity or abundance. Such a data structure could have biased the coefficients derived from the SEM, and we conducted data resampling to tackle this issue.

In the meta-regression depicted in Figure 6, we show the correlations between biodiversity responses and stressor intensity derived from our second-level meta-analysis. This is not accounting for the data structure (all the effect sizes on biodiversity or abundance are represented and included in the regression). The data resampling used in the SEM was no longer necessary because there were no repeated values of decomposition matching different decomposer diversity or abundance measurements in this univariate approach. This approach further complemented the information provided by the SEM by testing for the effect of moderators (such as study type and taxonomic groups). Therefore, the results from SEM and meta-regression differ because different datasets are used.

Furthermore, it is important to note that the magnitude of the relationships between Figure 5 (SEM) and Figure 6 (stressor intensity effect) are not comparable. Figure 5 shows the standardized coefficients, while Figure 6 shows unstandardized coefficients. The unstandardized coefficients from the SEM (Appendix 2—table 2) and the unstandardized coefficients of the stressor intensity relationships (Figure 6) are actually not that different from each other (with the SEM the slope was -0.10 (SE: 0.04, p = 0.02), while the meta-regression slope was = -0.05 (SE: 0.03, p = 0.11)). Again, these differences can be explained by the fact that different data underlie those results.

Finally, we re-tested the meta-regression between chemical stressor intensity and decomposer diversity responses, this time including an additional random effect of the litterbag (therefore addressing the data structure similarly as in the SEM). This approach yielded more similar results as the SEM (a significant stressor intensity effect on diversity responses: slope = -0.09, p=.009). We initially tested if including such an additional random effect improved the models. As this was not the case (Likelihood ratio test: 2.05, p = 0.15, ΔAIC = 0.05), we decided to keep the random effect structure minimal, thereby not including this additional litterbag random effect. The litterbag effect might have been confounded by the diversity metric effect in our models given that studies often reported different diversity metrics for the same litterbag sample.

We modified the main text to describe the potential reasons for this difference with more details:

“There was mixed support for a stressor intensity effect on decomposer diversity across the two approaches: decomposer diversity responses decreased with stressor intensity according to the SEM (Figure 5) but this trend was not significant according to the second level meta-analysis (Figure 6). […] The differences between the two approaches can be explained by the different data involved.”

We also discuss these findings in the Discussion section:

“These nutrient intensity patterns contrasted with the results for chemical stressors. […] Thus, negative responses to chemical stressors happened across the range of stressor intensity.”

In addition, it should be discussed to which extent the variable basis for the values used to characterize stressor intensity may be the reason for this finding. There is a relationship for nutrients, where the authors have a relatively consistent data source, whereas, for chemical stressors where the data sources vary, including the methods used to derive quality criteria, they found no relationship. Our suspicion is therefore that this is partly due to methodical reasons. To clarify, problems may arise if you choose a benchmark Y for stressor S1, but multiple benchmarks Zi where i from 1 to n for Stressor S2i that are not consistent with each other. Now imagine you find a relationship for one but not for the other with stressor intensity – this can simply be because you have variable benchmarks Zi and is not necessarily due to the stressor.

This is a very good point that we carefully considered prior to our analyses. We fully agree that discussing the variable standardization used is important, and we modified the text accordingly (see below). Chemical stressors in our meta-analysis include a wide range of chemicals (different metals, and pesticides). On the opposite, nutrient additions were far less variable, with only a hand full of nutrient types covered (NH_4_, NO_3_, NO_2_, PO_4_^3-^, total N and total P). It would be interesting to test the robustness of the results if we had only incorporated observations for which similar threshold sources could be derived. However, such a sensitivity analysis would not separate the “variable threshold effect” from the effect of certain type of chemicals versus others. This is because variable thresholds had to be used for particular types of pollutants (pesticide use, pesticide mixtures see below). Therefore, we are unsure whether the methodology to standardize stressor and nutrient intensity is the reason for the different relationships that we found. We rather think that the variable nature of chemical stressors versus nutrient enrichment, as defined in our analysis, is the reason for those differences.

The quality criteria values originated from consistent sources as much as possible. For chemicals, we mostly relied on two main authoritative sources: the European chemicals agency (ECHA) and the United States Environmental Protection Agency (USEPA). In our final datasets, we were able to retrieve quality criteria from those standard sources for 80% and 90% of the observations for diversity and abundance datasets, respectively. We preferably used ECHA and USEPA, because they derive quality criteria in a standardized way. Both are using similar approaches that rely on ecotoxicological data compiled for as many taxa as possible for different types of ecosystems. Importantly, these methodologies are consistent across aquatic and terrestrial realms, which was crucial for our approach.

However, using various sources for the threshold values was inevitable due to the high number of chemicals, as well as different ways authors reported stressor intensity in individual studies. We relied on authors’ statements for standardization in two main situations: terrestrial studies that reported pesticide intensity in terms of an application rate (n = 6 studies), and aquatic studies that tested the effect of mixture of a large number of pesticides, and reported pesticide levels as sums of toxic units (a method that is similar as what we used to standardize stressor levels, where the concentration of each pesticide is divided by standard ecotoxicological data before being summed across pesticides into an overall sum of toxic units) (n = 4). Because application rates and pesticide mixtures are not regulated by our two main authoritative sources, we relied on the authors’ statements (recommended application rates) and expert knowledge (safe levels for toxic units) to standardize stress intensity in those cases.

We agree with the reviewer that the nutrient-diversity dataset has a more homogeneous basis for standardization compared to the stressor-diversity dataset. However, we would like to stress the fact that our final datasets are all dominated by similar sources for standardizing stressors: thresholds from ECHA or USEPA for 80 and 90% of observations in the stressor-diversity and stressor-abundance datasets, respectively, and for nutrients, thresholds from WFD for 100 and 75% of observations in the nutrient-diversity and nutrient-abundance datasets, respectively. Therefore, it is possible that the results are little affected by the small number of observations for which quality criteria come from a different source than the main ones.

We modified the text to clarify our standardization approach in the Materials and methods section:

“We used consistent sources for the environmental quality criteria as much as possible. […] Despite this, the final datasets were all dominated by similar sources for standardizing stressor and nutrient intensity levels: thresholds from ECHA or USEPA for 80 and 90% of observations in the stressor-diversity and stressor-abundance datasets, respectively, and for nutrients, thresholds from WFD for 100 and 75% of observations in the nutrient-diversity and nutrient-abundance datasets, respectively.”

We further modified the Discussion section that now discuss the importance of standardization in our results:

“Such contrasting patterns between stressor and nutrient intensity effects may reflect the greater number of stressor types (different metals, pesticides, mixtures) covered by individual studies compared to nutrients. […] However, future studies focusing on stressor intensity effects across ecosystems would greatly benefit from coordinated efforts to derive quality criteria encompassing the vast and rapidly increasing number of chemical stressors (Wang et al., 2020).”

2) The stressor definition should be reconsidered. The authors define: "stressors (e.g., temperature, drought, chemicals) and resource enrichment (e.g., of CO_2_ or mineral nutrients)." From a theoretical perspective, the temperature would also be a niche-defining resource. From a practical perspective, it is confusing for the reader to contrast "chemical stressors" to CO_2_ and NO_3_, the latter of which are clearly chemicals and often anthropogenic. Then both are "environmental change drivers", which makes the terminology even more complicated for the reader. Why not just use stressors and environmental change drivers, which have different niche-defining properties? Some are unimodal (nutrients, temperature), others exhibit a threshold relationship and others may be log-linear.

We agree that the previous wording may have been confusing. Chemical stressors are environmental change drivers (see e.g. Bernhardt et al., 2017 Frontiers in Ecology and the Environment), but contrarily to nutrients, they cannot be consumed. This distinction (being consumed by the focal populations or not) is key to our conceptual framework. Whether or not these variables have implications for niche partitioning is not part of our framework. In that sense, temperature cannot be consumed and therefore does categorize as a stressor in our framework. Therefore, we would like to maintain the adopted terminology. However, we have now changed the wording in the Introduction to define our two main categories in a clearer way, but to also mention potential caveats with any grouping approach. The text now reads:

“We postulate that there are two main categories of environmental change: stressors and resource shifts. While stressors cannot be consumed, and act as conditions that alter growth rates (e.g., temperature, drought, chemical stressors), resources are by definition consumed (e.g., CO_2_ or mineral nutrients), which has important implications for how they should enter theory (De Laender, 2018; Chase and Leibold, 2003).”

We further added a statement in the Discussion highlighting that additional groupings of drivers could be addressed by future work focusing on a greater number of drivers:

“Finally, although our comparison of stressors versus resources allowed us to test a clear concept, any kind of grouping in ecological studies may mask some of the variation within the categories, and future studies may be interested in different categories. […] This would also allow to test additional groupings of drivers and ecological concepts unifying stressors and resources (De Laender, 2018; Harley et al., 2017).”